# Cell-autonomous role of leucine-rich repeat kinase in the protection of dopaminergic neuron survival

**Jongkyun Kang[1†], Guodong Huang[1†], Long Ma[1], Youren Tong[1], Anu Shahapal[1], Phoenix Chen[1], Jie Shen[1,2]***

[1]Department of Neurology, Brigham and Women's Hospital, Boston, United States; [2]Program in Neuroscience, Harvard Medical School, Boston, United States

**\*For correspondence:**
jshen@bwh.harvard.edu

[†]These authors contributed equally to this work

**Competing interest:** The authors declare that no competing interests exist.

**Abstract** Mutations in leucine-rich repeat kinase 2 (LRRK2) are the most common genetic cause of Parkinson's disease (PD). However, whether LRRK2 mutations cause PD and degeneration of dopaminergic (DA) neurons via a toxic gain-of-function or a loss-of-function mechanism is unresolved and has pivotal implications for LRRK2-based PD therapies. In this study, we investigate whether *Lrrk2* and its functional homolog *Lrrk1* play a cell-intrinsic role in DA neuron survival through the development of DA neuron-specific *Lrrk* conditional double knockout (cDKO) mice. Unlike *Lrrk* germline DKO mice, DA neuron-restricted *Lrrk* cDKO mice exhibit normal mortality but develop age-dependent loss of DA neurons, as shown by the progressive reduction of DA neurons in the substantia nigra pars compacta (SNpc) at the ages of 20 and 24 months. Moreover, DA neurodegeneration is accompanied with increases in apoptosis and elevated microgliosis in the SNpc as well as decreases in DA terminals in the striatum, and is preceded by impaired motor coordination. Taken together, these findings provide the unequivocal evidence for the cell-intrinsic requirement of LRRK in DA neurons and raise the possibility that LRRK2 mutations may impair its protection of DA neurons, leading to DA neurodegeneration in PD.

## eLife assessment

This current revision builds on observations in validated conditional double KO (cDKO) mice for LRRK1 and LRRK2 that will be **useful** for the field, given that LRRK2 is widely expressed in the brain and periphery, and many divergent phenotypes have been attributed previously to LRRK2 expression. The article presents **solid** data demonstrating that it is the loss of LRRK1 and LRRK2 expression within the SNpc DA cells that is not well tolerated as it was previously unclear from past work whether neurodegeneration in the LRRK double knock out (DKO) was cell autonomous or the result of loss of LRRK1/LRRK2 expression in other types of cells. Future studies may pursue the biochemical mechanisms underlying the reason for the apoptotic cells noted in this study, as here, the LRRK1/LRRK2 KO mice did not replicate the dramatic increase in autophagic vacuole numbers previously noted in the germline global LRRK1/LRRK2 KO mice.

## Introduction

Parkinson's disease (PD) is the most common movement disorder and is characterized by the progressive loss of dopaminergic (DA) neurons in the substantia nigra pars compacta (SNpc). Dominantly inherited missense mutations in the *leucine-rich repeat kinase 2* (*Lrrk2*) gene are the most common cause of both familial and sporadic PD, highlighting the importance of LRRK2 in PD pathogenesis (*Paisán-Ruíz et al., 2004*; *Shen, 2004*; *Zimprich et al., 2004*; *Gilks et al., 2005*; *Lesage*

*et al., 2007*; *Kluss et al., 2019*; *Shu et al., 2019*; *Mata et al., 2005a*; *Mata et al., 2005b*; *Zabetian et al., 2005*; *Hatano et al., 2014*; *Takanashi et al., 2018*; *Nichols et al., 2005*; *Di Fonzo et al., 2005*; *Hernandez et al., 2005*; *Kachergus et al., 2005*). LRRK2 is a large protein of 2527 amino acid residues containing multiple functional domains, including several leucine-rich repeats (LRRs), a GTPase-like domain of Ras-of-complex (Roc), a C-terminal of Roc (COR) domain, and a serine/threonine MAPKKK-like kinase domain. LRRK1, a homolog of LRRK2, belongs to the evolutionarily conserved Roco protein family and contains similar LRRs, Roc, COR, and kinase domains (*Bosgraaf and Van Haastert, 2003*; *Marín, 2006*; *Marín, 2008*). LRRK proteins are broadly expressed, with LRRK2 being most abundant in the kidney (*Biskup et al., 2007*). While most PD mutations are found in LRRK2, rare variants in the Roc, COR, and kinase domains of LRRK1 have been reported and may be associated with PD (*Schulte et al., 2014*). Furthermore, the Roc-COR domain of LRRK2 forms dimers and exhibits conventional Ras-like GTPase properties, and the R1441/C/G/H and I1371V mutations destabilize dimer formation and decrease GTPase activity (*Deng et al., 2008*; *Mills et al., 2018*). Recent high-resolution cryoEM structural studies of full-length LRRK2 demonstrated its existence as dimers and pathogenic mutations such as R1441/C/G/H and Y1699I at the Roc-COR interface, whereas the G2910S mutant is structurally similar to the wild-type LRRK2 (*Myasnikov et al., 2021*).

Previous genetic studies demonstrated that LRRK2 plays essential roles in the autophagy-lysosomal pathway (*Tong et al., 2010*; *Tong et al., 2012*; *Herzig et al., 2011*; *Tian et al., 2021*). Consistent with high levels of LRRK2 expression in kidneys, *Lrrk2*[-/-] mice develop age-dependent phenotypes in the kidney, including autophagy-lysosomal impairments and increases in α-synuclein, apoptosis, and inflammatory responses (*Tong et al., 2010*; *Tong et al., 2012*; *Tong and Shen, 2012*). It has also been reported that LRRK2 is important for maintaining lung homeostasis, and *Lrrk2* deficiency results in impaired autophagy in alveolar type II epithelial cells (*Tian et al., 2021*). It was proposed that the lack of brain phenotypes in *Lrrk2*[-/-] mice might be due to the presence of LRRK1, which could compensate functionally in the absence of LRRK2 (*Tong et al., 2010*; *Tong et al., 2012*). Indeed, *Lrrk* double knockout (DKO) mice develop an age-dependent, progressive loss of DA neurons in the SNpc, beginning at 14 months of age (*Giaime et al., 2017*). However, *Lrrk* DKO mice also exhibit lower body weight and earlier mortality, raising the possibility that DA neurodegeneration in aged *Lrrk* germline DKO mice may be secondary to poor health.

In this study, we investigate whether LRRK2 and its functional homolog LRRK1 play an essential, intrinsic role in DA neurons through the development of DA neuron-specific *Lrrk* conditional DKO (cDKO) mice using the *Slc6a3-Cre* knockin (KI) allele, in which Cre recombinase is expressed under the control of the endogenous promoter of the dopamine transporter gene *Slc6a3*. We first generated and confirmed floxed *Lrrk1* and *Lrrk2* mice, and then crossed them with the *CMV-Cre* deleter to create germline deletions of the floxed *Lrrk1* and *Lrrk2* alleles, followed by northern and western analyses to confirm the absence of *Lrrk1* and *Lrrk2* mRNAs, as well as LRRK1 and LRRK2 proteins in the respective homozygous deleted mutant mice. We also crossed a GFP reporter mouse with *Slc6a3-Cre* KI mice and confirmed that Cre-mediated recombination occurs in most, if not all, DA neurons in the SNpc and is restricted to DA neurons. We then crossed these thoroughly validated floxed *Lrrk1* and *Lrrk2* mice with *Slc6a3-Cre* KI mice to generate DA neuron-restricted *Lrrk* cDKO mice and further confirmed the reduction of LRRK1 and LRRK2 in dissected ventral midbrains of *Lrrk* cDKO mice. While DA neuron-restricted *Lrrk* cDKO mice of both sexes exhibit normal mortality and body weight, they develop age-dependent loss of DA neurons in the SNpc, as demonstrated by the progressive reduction of TH+ DA neurons or NeuN+ neurons in the SNpc of cDKO mice at the ages of 20 and 24 months but not at 15 months. Moreover, DA neurodegeneration is accompanied with increases in apoptotic DA neurons and elevated microgliosis in the SNpc as well as decreases in DA terminals in the striatum, and is preceded by impaired motor coordination. Interestingly, quantitative electron microscopy (EM) analysis showed a similar number of electron-dense vacuoles in SNpc neurons of *Lrrk* cDKO mice relative to controls, in contrast to age-dependent increases in vacuoles in SNpc neurons of *Lrrk* germline DKO mice (*Giaime et al., 2017*; *Huang et al., 2022*). These findings provide the unequivocal evidence for the importance of LRRK in DA neurons and raise the possibility that LRRK2 mutations may impair this crucial physiological function, leading to DA neurodegeneration in PD.

## Results

### Generation and molecular characterization of the floxed and deleted *Lrrk1* and *Lrrk2* alleles

LRRK2 and its homolog LRRK1 share several functional (LRRs, GTPase Roc, COR, and kinase) domains (*Figure 1A*). To investigate the intrinsic role of LRRK in DA neurons, we generated floxed *Lrrk1* (*Lrrk1*$^{F/F}$) and floxed *Lrrk2* (*Lrrk2*$^{F/F}$) mice, which permit deletions of *Lrrk1* and *Lrrk2* selectively in DA neurons by the *Slc6a3-Cre* KI allele (*Slc6a3*$^{Cre/+}$), in which Cre recombinase is expressed under the control of the endogenous promoter of the dopamine transporter gene *Slc6a3* (*Bäckman et al., 2006*). We introduced two loxP sites in introns 26 and 29 of *Lrrk1* through homologous recombination and site-specific recombination by FLP recombinase to remove the positive selection *Pgk-neo* cassette, which is flanked by two FRT sites (*Figure 1B and C*; *Supplementary file 1*; *Figure 1—figure supplement 1*). The embryonic stem (ES) cells carrying the targeted allele or the floxed allele were identified and validated by Southern analysis using the 5' and 3' external probes as well as the *neo* probe (*Figure 1D*, *Figure 1—figure supplement 2*). The validated ES cells carrying the *Lrrk1*$^{F/+}$ allele were injected into mouse blastocysts to generate *Lrrk1*$^{F/+}$ mice, which were further confirmed by Southern using the 5' and 3' external probes (*Figure 1D*). In the presence of Cre recombinase, the floxed *Lrrk1* genomic region containing part of intron 26 (1288 bp), exons 27–29, which encode the kinase domain, and part of intron 29 (1023 bp) is deleted (*Figure 1C*), and the removal of exons 27–29 (625 bp) results in a frameshift of downstream codons.

The *Lrrk2* targeting vector contains the 5' homologous region, a loxP site 1768 bp upstream of the transcription initiation site, exons 1–2, the *Pgk-neo* cassette flanked by two loxP sites and two FRT sequences, and the 3' homologous region (*Figure 1B and E*; *Supplementary file 2*; *Figure 1—figure supplement 3*). The ES cell clones carrying the targeted allele were identified and validated by Southern analysis using the 5' external probe (*Figure 1F*, *Figure 1—figure supplement 4*) and genomic PCR followed by sequencing. Mice carrying the *Lrrk2* targeted allele were further verified by Southern analysis using the 5' and 3' external probes (*Figure 1—figure supplement 4*) as well as the *neo* probe and were then crossed with the *Actin-FLP* mice (*Rodríguez et al., 2000*) to remove the *Pgk-neo* cassette flanked by two FRT sites to generate *Lrrk2*$^{F/+}$ mice (*Figure 1E*). The resulting *Lrrk2*$^{F/+}$ mice were confirmed by Southern analysis using the 5' and 3' external probes (*Figure 1F*). In the presence of Cre recombinase, the floxed *Lrrk2* region containing the promoter and exons 1–2 are deleted, likely resulting in a null allele, as we previously targeted a very similar region (~2.5 kb upstream of the transcription initiation site and exons 1–2) to generate germline deletion of *Lrrk2* (*Tong et al., 2010*).

DA neurons in the SNpc are a small neuronal population embedded in the ventral midbrain, making it difficult to confirm whether DA neuron-specific deletions of the floxed *Lrrk1* and *Lrrk2* regions result in null alleles. We previously generated three independent *Lrrk1* knockout (KO) mice, and only one KO line (line 2) represents a *Lrrk1*-null allele (*Giaime et al., 2017*), whereas deletion of *Lrrk2* promoter region and exons 1–2 resulted in a *Lrrk2*-null allele (*Tong et al., 2010*). We therefore generated germline deleted (Δ/Δ) *Lrrk1* (*Lrrk1*$^{Δ/Δ}$) and *Lrrk2* (*Lrrk2*$^{Δ/Δ}$) mice from *Lrrk1*$^{F/F}$ and *Lrrk2*$^{F/F}$ mice, respectively, by crossing them to germline deleter *CMV-Cre* mice (*Schwenk et al., 1995*).

We then performed northern analysis of *Lrrk1* using both an upstream probe specific for exons 2–3 and a downstream probe specific for exons 27–29 (*Figure 1G*, *Figure 1—figure supplements 5 and 6*). Because of the low expression level of *Lrrk1* mRNA and the relative abundance of *Lrrk1* mRNA in the lung (*Biskup et al., 2007*), we enriched polyA+ RNA from the lung of the mice carrying homozygous germline deleted *Lrrk1*$^{Δ/Δ}$ alleles derived from *Lrrk1*$^{F/F}$ alleles. Using the exons 2–3 probe, the *Lrrk1* transcripts in wild-type mice are of the expected size of ~7.4 kb, whereas the detected *Lrrk1* transcripts in *Lrrk1*$^{Δ/Δ}$ mice are smaller and are expressed at lower levels, consistent with the deletion of exons 27–29 (625 bp), resulting in a frameshift of downstream codons and the likely degradation of the truncated *Lrrk1* mRNA (*Figure 1G*; *Figure 1—figure supplement 6*). Using a probe specific for exons 27–29, there is no *Lrrk1* transcript in *Lrrk1*$^{Δ/Δ}$ mice (*Figure 1G*, *Figure 1—figure supplement 6*). Extensive RT-PCR analysis of total RNA isolated from the kidney, brain, and lung of *Lrrk1*$^{Δ/Δ}$ mice using an exon 32-specific primer for RT and exon-specific primer sets for PCR (e.g., exons 4–8, 11–17, 20–25, and 25–31), followed by sequencing confirmation of the PCR products, indicated normal *Lrrk1* splicing in *Lrrk1*$^{F/F}$ mice and the lack of *Lrrk1* exons 27–29 in *Lrrk1*$^{Δ/Δ}$ mice (*Figure 1—figure supplement 7*).

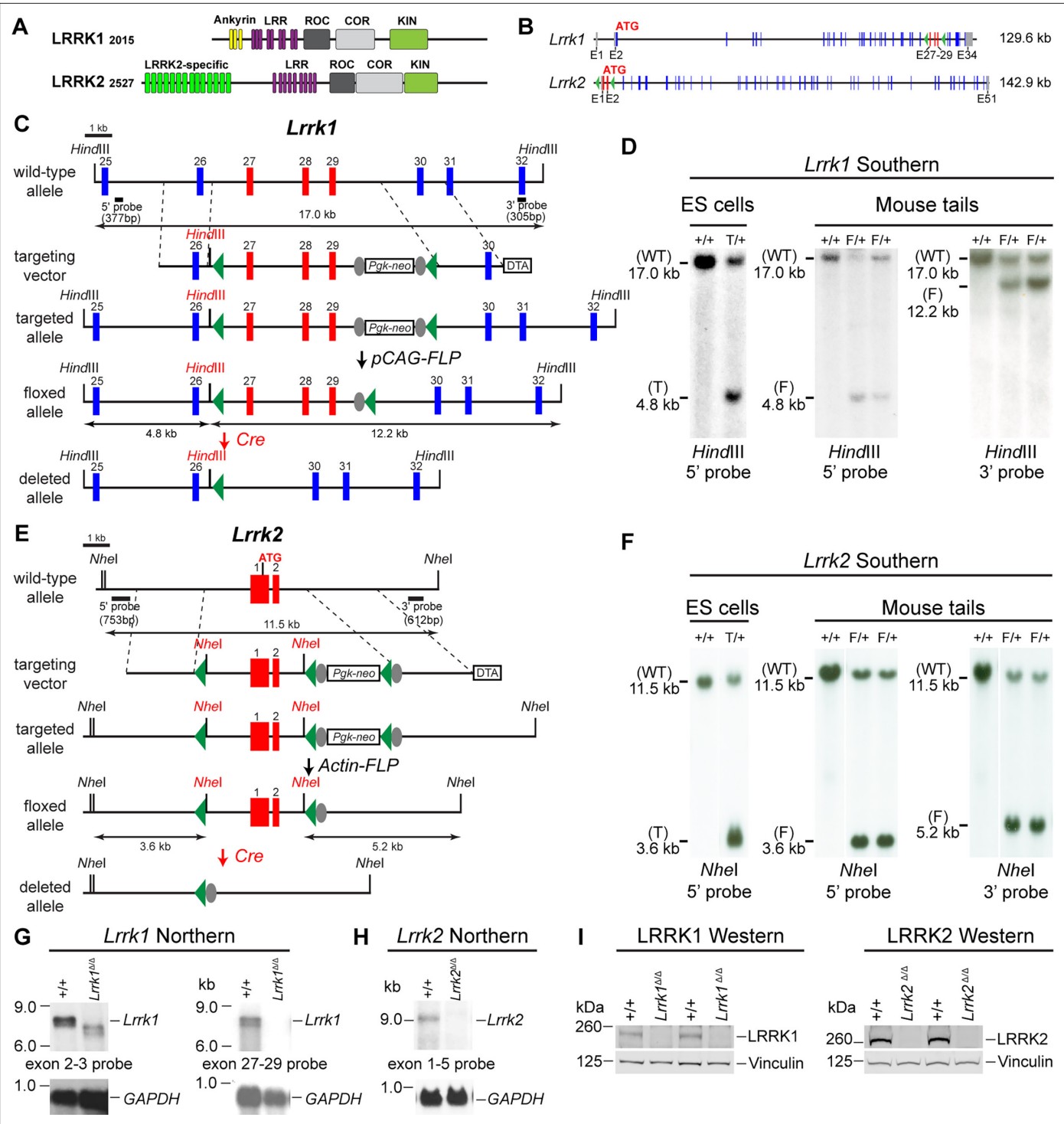

**Figure 1.** Generation and characterization of floxed and deleted *Lrrk1* and *Lrrk2* alleles. (**A**) Schematic illustrations of human LRRK1 and LRRK2 proteins showing similar functional domains. LRRK1₂₀₁₅ protein is derived from exons 2–34 (Ensembl Genome Database: ENSG00000154237). LRRK2₂₅₂₇ protein is derived from exons 1–51 (ENSG00000188906). LRR: leucine-rich repeats; Roc: Ras-of-complex; COR: C-terminal of Roc; KIN: kinase domain. (**B**) Schematic illustrations of the gene structures of mouse *Lrrk1* and *Lrrk2*. The boxes in blue are exons that encode the LRRK1 and LRRK2 proteins, and the gray boxes represent the 5' and 3' UTRs. The exons are not drawn in scale. The start codon ATG is in exon 2 of *Lrrk1* and exon 1 of *Lrrk2*. The exons 27–29 of *Lrrk1* and the promoter/exons 1–2 of *Lrrk2* are flanked with loxP sites (green arrowheads). (**C**) Targeting strategy for the generation of the targeted, floxed, and deleted *Lrrk1* alleles. The red boxes represent the targeted exons 27–29, and the blue boxes represent the untargeted exons. The locations and sizes of the 5' and 3' external probes are shown. The targeting vector contains the 5' and 3' homologous regions (marked by dashed lines) and the middle region (from intron 26 to intron 29), which includes a loxP site (green arrowhead) in intron 26 (1288 bp upstream of exon

*Figure 1 continued*

27) and the *Pgk-neo* selection cassette flanked by two FRT (FLP recognition target) sequences (gray circles) followed by another loxP site in intron 29 (1023 bp downstream of exon 29). A negative selection cassette encoding diphtheria toxin fragment A (DTA) is also included in the targeting vector to reduce embryonic stem (ES) cells bearing randomly inserted targeting vectors. ES cells carrying the correctly targeted *Lrrk1* allele were transfected with *pCAG-FLP* to remove the *Pgk-neo* cassette and generate the floxed *Lrrk1* allele. Floxed *Lrrk1* mice were bred with *CMV-Cre* transgenic mice to generate germline deleted *Lrrk1* mice. Detailed strategy for generating targeting vector and DNA sequence of floxed *Lrrk1* allele can be found in ***Figure 1— figure supplement 1*** and ***Supplementary file 1***, respectively. (**D**) Southern analysis of the targeted and floxed *Lrrk1* alleles. Genomic DNA from ES cells or mouse tails was digested with *Hind*III and hybridized with the 5' or 3' external probe. For the 5' probe, the resulting 17.0 kb and 4.8 kb bands represent the wild-type (WT) and the targeted (T) or floxed (F) alleles, respectively. For the 3' probe, the resulting 17.0 kb and 12.2 kb bands represent the WT and the floxed alleles, respectively. Detailed Southern strategy can be found in ***Figure 1—figure supplement 2***. (**E**) Targeting strategy for the generation of the targeted, floxed, and deleted *Lrrk2* alleles. The red boxes represent *Lrrk2* exons 1 and 2, and the start codon ATG resides in exon 1. The locations and sizes of the 5' and 3' external probes are shown. The targeting vector contains the 5' and 3' homologous regions (marked by dashed lines) and the middle region (from the promoter to intron 2), which includes a loxP site (green arrowhead) upstream (1768 bp) of the transcription initiation site and the *Pgk-neo* selection cassette flanked by two FRT sequences (gray circles) and two loxP sites (green arrowheads) in intron 2 (878 bp downstream of exon 2). A negative selection cassette encoding DTA is also included in the targeting vector. Mice carrying the correctly targeted *Lrrk2* allele were crossed with *Actin-FLP* deleter mice to generate floxed *Lrrk2* mice, which were bred with *CMV-Cre* transgenic mice to generate germline deleted *Lrrk2* mice. Detailed strategy for generating targeting vector and the DNA sequence of the floxed *Lrrk2* allele can be found in ***Figure 1—figure supplement 3***, respectively. (**F**) Southern analysis of the targeted and floxed *Lrrk2* alleles. Genomic DNA from ES cells or mouse tails was digested with *Nhe*I and hybridized with the 5' or 3' external probe. For the 5' probe, the resulting 11.5 kb and 3.6 kb bands represent the wild-type (WT) and the targeted (T) or floxed (F) alleles, respectively. For the 3' probe, the resulting 11.5 kb and 5.2 kb bands represent the WT and the floxed *Lrrk2* alleles, respectively. Detailed Southern strategy can be found in ***Figure 1—figure supplement 4***. (**G**) Northern analysis of poly(A)+RNA prepared from the lung of *Lrrk1*$^{\Delta/\Delta}$ mice carrying homozygous germline deleted (Δ/Δ) *Lrrk1* alleles derived from *Lrrk1*$^{F/F}$ mice using the cDNA probe of exons 2–3 (left) and exons 27–29 (right). Using the upstream exons 2–3 probe, the *Lrrk1* transcripts in wild-type mice are the expected size of ~7.4 kb, whereas the detected *Lrrk1* transcripts in *Lrrk1*$^{\Delta/\Delta}$ mice are truncated, consistent with the deletion of exons 27–29 (625 bp), which results in a frameshift, and are expressed at lower levels, likely due to nonsense-mediated degradation of the truncated *Lrrk1* mRNA. Using a probe specific for exons 27–29, there is no *Lrrk1* transcript in *Lrrk1*$^{\Delta/\Delta}$ mice, as expected. Both blots were hybridized with a *GAPDH* probe as loading controls. Detailed northern strategy and full-size blots are included in ***Figure 1—figure supplements 5 and 6***, respectively. Extensive RT-PCR analysis of *Lrrk1* transcripts in *Lrrk1*$^{\Delta/\Delta}$ mice is shown in ***Figure 1— figure supplement 7***. (**H**) Northern analysis of total RNA prepared from the neocortex of *Lrrk2*$^{\Delta/\Delta}$ mice carrying homozygous germline *Lrrk2* deleted (Δ/Δ) alleles using the cDNA probe of exons 1–5 shows the absence of *Lrrk2* transcripts. The blot was hybridized with a *GAPDH* probe as a loading control. The full-size blot is included in ***Figure 1—figure supplement 8***. RT-PCR analysis of *Lrrk2* transcripts in *Lrrk2*$^{\Delta/\Delta}$ mice is shown in ***Figure 1— figure supplement 9***. (**I**) Left: western analysis of wild-type (+/+) and homozygous *Lrrk1*$^{\Delta/\Delta}$ (Δ/Δ) brains shows the absence of LRRK1 protein. Right: western analysis of the neocortex of wild-type (+/+) and homozygous *Lrrk2*$^{\Delta/\Delta}$ (Δ/Δ) mice shows the absence of LRRK2 protein. Vinculin was used as a loading control. Full-size blots can be found in ***Figure 1—source data 1***.

The online version of this article includes the following source data and figure supplement(s) for figure 1:

**Source data 1.** Full-size western blots of *Lrrk1*$^{\Delta/\Delta}$ and *Lrrk2*$^{\Delta/\Delta}$ brains.

**Figure supplement 1.** Genomic DNA sequence of the floxed *Lrrk1* allele.

**Figure supplement 2.** Southern strategy for the targeted and floxed *Lrrk1* alleles.

**Figure supplement 3.** Genomic DNA sequence of the floxed *Lrrk2* allele.

**Figure supplement 4.** Southern analysis of the targeted and floxed *Lrrk2* alleles.

**Figure supplement 5.** Northern strategy for *Lrrk1* mRNA.

**Figure supplement 6.** Northern analysis of *Lrrk1* mRNA in *Lrrk1*$^{\Delta/\Delta}$ mice.

**Figure supplement 7.** RT-PCR analysis of the deleted *Lrrk1* allele.

**Figure supplement 8.** Northern analysis of *Lrrk2* mRNA in *Lrrk2*$^{F/F}$ and *Lrrk2*$^{\Delta/\Delta}$ mice.

**Figure supplement 9.** RT-PCR analysis of the deleted *Lrrk2* allele.

Similarly, northern analysis of *Lrrk2* using a probe specific for exons 1–5 and RT-PCR followed by sequencing confirmed the absence of *Lrrk2* mRNA in *Lrrk2*$^{\Delta/\Delta}$ brains and normal *Lrrk2* transcripts in *Lrrk2*$^{F/F}$ brains (***Figure 1H***, ***Figure 1—figure supplements 8 and 9***). Furthermore, western analysis confirmed the absence of LRRK1 and LRRK2 proteins in the brain of *Lrrk1*$^{\Delta/\Delta}$ and *Lrrk2*$^{\Delta/\Delta}$ mice, respectively (***Figure 1I***). Taken together, our northern, RT-PCR followed by sequencing, and western analyses demonstrated that deletion of the floxed *Lrrk1* and *Lrrk2* alleles results in null mutations. Thus, floxed *Lrrk1* and *Lrrk2* alleles can be used to generate DA neuron-specific *Lrrk* cDKO mice.

## Generation and molecular characterization of DA neuron-specific *Lrrk* cDKO mice

To generate DA neuron-specific *Lrrk* cDKO mice, we used *Slc6a3$^{Cre/+}$* KI mice, which express Cre recombinase under the control of the dopamine transporter gene *Slc6a3* (*Bäckman et al., 2006*). To confirm if Cre-mediated recombination occurs broadly and specifically in DA neurons of the SNpc, we crossed *Slc6a3$^{Cre/+}$* mice with the *Rosa26$^{CAG-LSL-ZsGreen1}$* reporter mouse (*Madisen et al., 2010*). Upon Cre expression, Cre recombinase removes the floxed 'stop' cassette, resulting in the expression of EGFP. We found that Cre-mediated recombination (GFP+) occurs in TH+ cells in the SNpc (*Figure 2A*). Quantification of GFP+ and/or TH+ cells in the SNpc showed that 99% of TH+ DA neurons are also GFP+, demonstrating that Cre-mediated recombination takes place in essentially all DA neurons in the SNpc (*Figure 2B*).

Having confirmed *Lrrk1$^{F/F}$* and *Lrrk2$^{F/F}$* mice as well as *Slc6a3$^{Cre/+}$* mice, we then bred them together to generate *Lrrk* cDKO mice (*Lrrk1$^{F/F}$; Lrrk2$^{F/F}$; Slc6a3$^{Cre/+}$*), which were further bred with *Lrrk1$^{F/F}$; Lrrk2$^{F/F}$* mice to generate cDKO and littermate controls (*Lrrk1$^{F/F}$; Lrrk2$^{F/F}$*). It was previously reported that germline *Lrrk* DKO mice failed to gain weight as they aged (*Giaime et al., 2017*). However, DA neuron-specific *Lrrk* cDKO and littermate control mice have similar body and brain weights at the ages of 2–24 months (*Figure 2C and D*). Western analysis showed a significant reduction of LRRK1 and LRRK2 proteins in the dissected ventral midbrain but not in the cerebral cortex of DA neuron-specific *Lrrk* cDKO mice at 2–3 months of age, relative to littermate controls (*Figure 2E and F*), further validating these DA neuron-specific *Lrrk* cDKO mice.

## Age-dependent loss of TH+ DA neurons in the SNpc of *Lrrk* cDKO mice

To determine whether the inactivation of LRRK selectively in DA neurons of the SNpc affects their survival, we performed TH immunostaining and quantified TH+ DA neurons in the SNpc of *Lrrk* cDKO mice and littermate controls using stereological methods. The morphology of TH+ DA neurons in *Lrrk* cDKO mice at the ages of 15, 20, and 24 months appears normal (*Figure 3A*). Quantification of TH+ neurons in the SNpc using serial sections emcompassing the entire SNpc revealed that the number of DA neurons in the SNpc at the age of 15 months is similar between cDKO mice (10,000 ± 141) and littermate controls (10,077 ± 310; F$_{1,46}$ = 16.59, p=0.0002, two-way ANOVA with Bonferroni's post hoc multiple comparisons, p>0.9999; *Figure 3B*). However, at the age of 20 months, the number of TH+ neurons in the SNpc of cDKO mice (8948 ± 273) is significantly reduced compared to controls (10,244 ± 220; p=0.0041), and is further decreased at the age of 24 months (control: 9675 ± 232, cDKO: 8188 ± 452; p=0.0010; *Figure 3B*). Similar genotypic differences were observed in an independent quantification by another investigator, also conducted in a genotype-blind manner, using the fractionator and optical dissector to randomly sample 25% area of the SNpc (*Figure 3—figure supplement 1*).

We further performed TH and NeuN double immunostaining of *Lrrk* cDKO and control mice at 24 months of age (*Figure 3C*). Quantification of NeuN+ and TH+/NeuN+ cells in the SNpc using serial sections encompassing the entire SNpc showed that the number of NeuN+ neurons is also significantly reduced in the SNpc of cDKO mice (17,923 ± 813) compared to controls (21,907 ± 469, p=0.0006, Student's *t*-test; *Figure 3D*). The number of TH+/NeuN+ cells is also lower in the SNpc of *Lrrk* cDKO mice (10,500 ± 644) compared to control mice (14,102 ± 310, p=0.0001; *Figure 3D*). These data indicate that the reduction in TH+ cells is not due to decreases in TH expression in DA neurons, but rather a result of the loss of DA neuron cell bodies in the SNpc of *Lrrk* cDKO mice.

We further evaluated apoptosis in the SNpc of *Lrrk* cDKO and littermate controls at the age of 24 months using an antibody specific for active Caspase-3 to label apoptotic cells. We observed increases in apoptotic DA neurons, labeled by active Caspase-3+ and TH+ immunoreactivity, in the SNpc of *Lrrk* cDKO mice (*Figure 4A*). Quantification of active Caspase-3+/TH+ apoptotic DA neurons shows a significant increase in the SNpc of *Lrrk* cDKO mice (323 ± 38) compared to controls (157 ± 8, p=0.0004, Student's *t*-test; *Figure 4B*). These results further support that LRRK plays an intrinsic role in the survival of DA neurons in the SNpc during aging.

## Age-dependent loss of TH+ DA terminals in the striatum of *Lrrk* cDKO mice

To determine whether loss of DA neurons in the SNpc is accompanied with loss of DA terminals in the striatum, we performed TH immunostaining and quantified TH immunoreactivity in the striatum

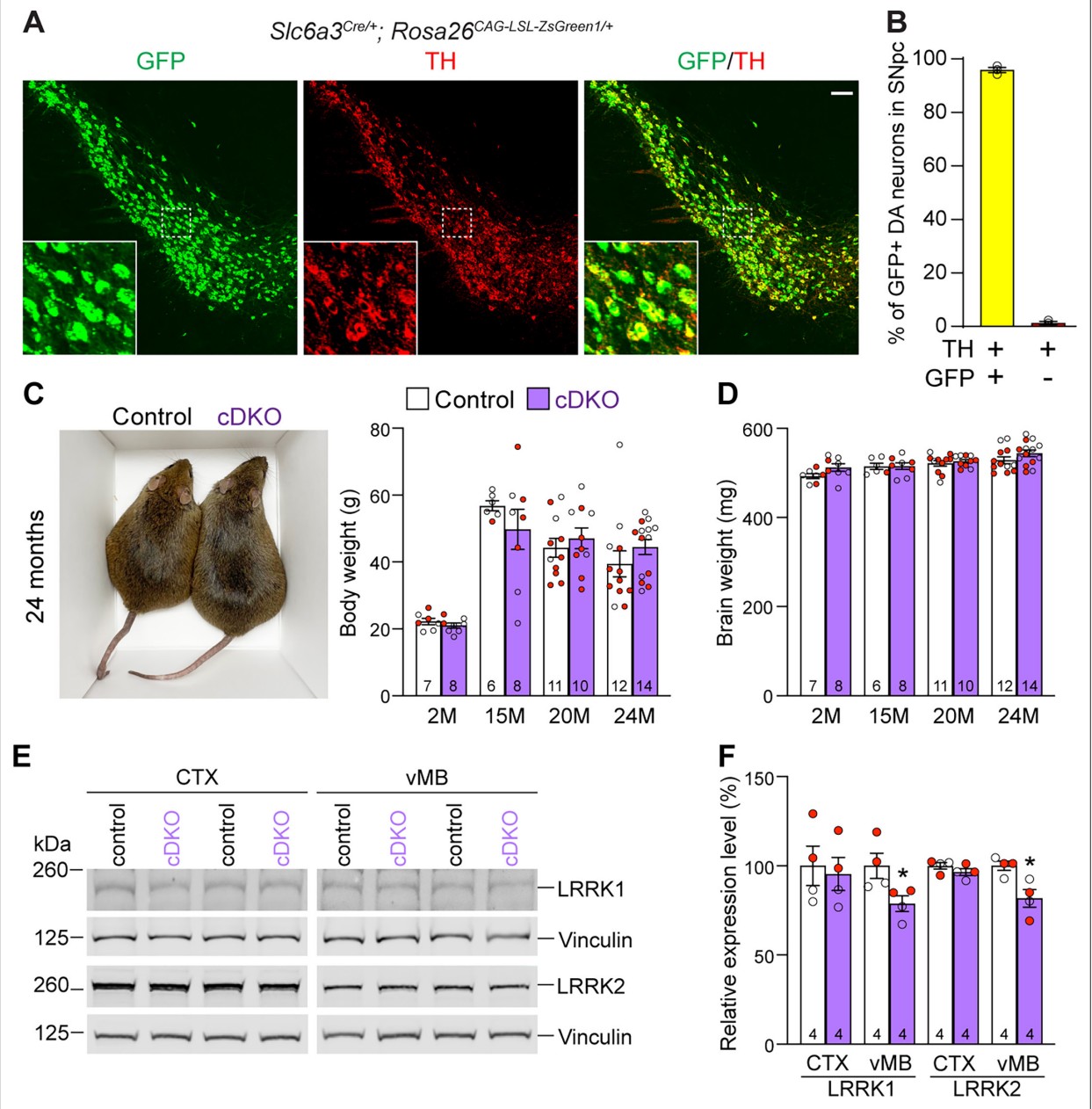

**Figure 2.** Generation and characterization of dopaminergic (DA) neuron-specific *Lrrk* conditional double knockout (cDKO) mice. (**A**) Immunostaining of GFP and/or TH in the SNpc of *Slc6a3^{Cre/+}; Rosa26^{CAG-LSL-ZsGreen1/+}* mice at 2 months of age. Cre recombinase is expressed under the control of the *Slc6a3* endogenous promoter and removes the floxed 'stop' cassette, resulting in the expression of EGFP under the control of the ubiquitous *CAG* promoter. (**B**) Quantification of GFP+/TH+ and TH+ cells shows that 99% of TH+ DA neurons (722 ± 46 TH+ cells) in the SNpc are also GFP+ (713 ± 46 cells), indicating that *Slc6a3-Cre* mediated recombination occurs in essentially all TH+ DA neurons. N = 3 mice, three comparable sections per hemisphere, 320 μm apart. (**C**) Similar body weight between *Lrrk* cDKO mice and littermate controls at all ages examined ($F_{1,68}$ = 0.001310, p=0.9712; 2M, 20M: p>0.9999; 15M: p=0.7857, 25M: p=0.8084, two-way ANOVA with Bonferroni's post hoc multiple comparisons). (**D**) Similar brain weight between *Lrrk* cDKO and control mice ($F_{1,68}$ = 3.603, p=0.0619; 2M: p=0.3893; 15M: p>0.9999; 20M: p>0.9999; 25M: p=0.3223, two-way ANOVA with Bonferroni's post hoc multiple comparisons). (**E**) Western analysis of LRRK1 and LRRK2 proteins in the dissected cerebral cortex (CTX) and ventral midbrain (vMB) of *Lrrk* cDKO and littermate controls at 2 months of age. (**F**) Quantification shows significant decreases in LRRK1 and LRRK2 in the dissected ventral midbrain of *Lrrk* cDKO mice (LRRK1, p=0.0432; LRRK2, p=0.0162, Student's *t*-test), compared to controls, but not in the dissected cortex of cDKO mice (LRRK1: p=0.7648; LRRK2: p=0.2325). The number in the column indicates the number of mice used in the study. Red-filled and open circles represent data obtained from individual male and female mice, respectively. All data are expressed as mean ± SEM. *p<0.05. Scale bar: 100 μm.

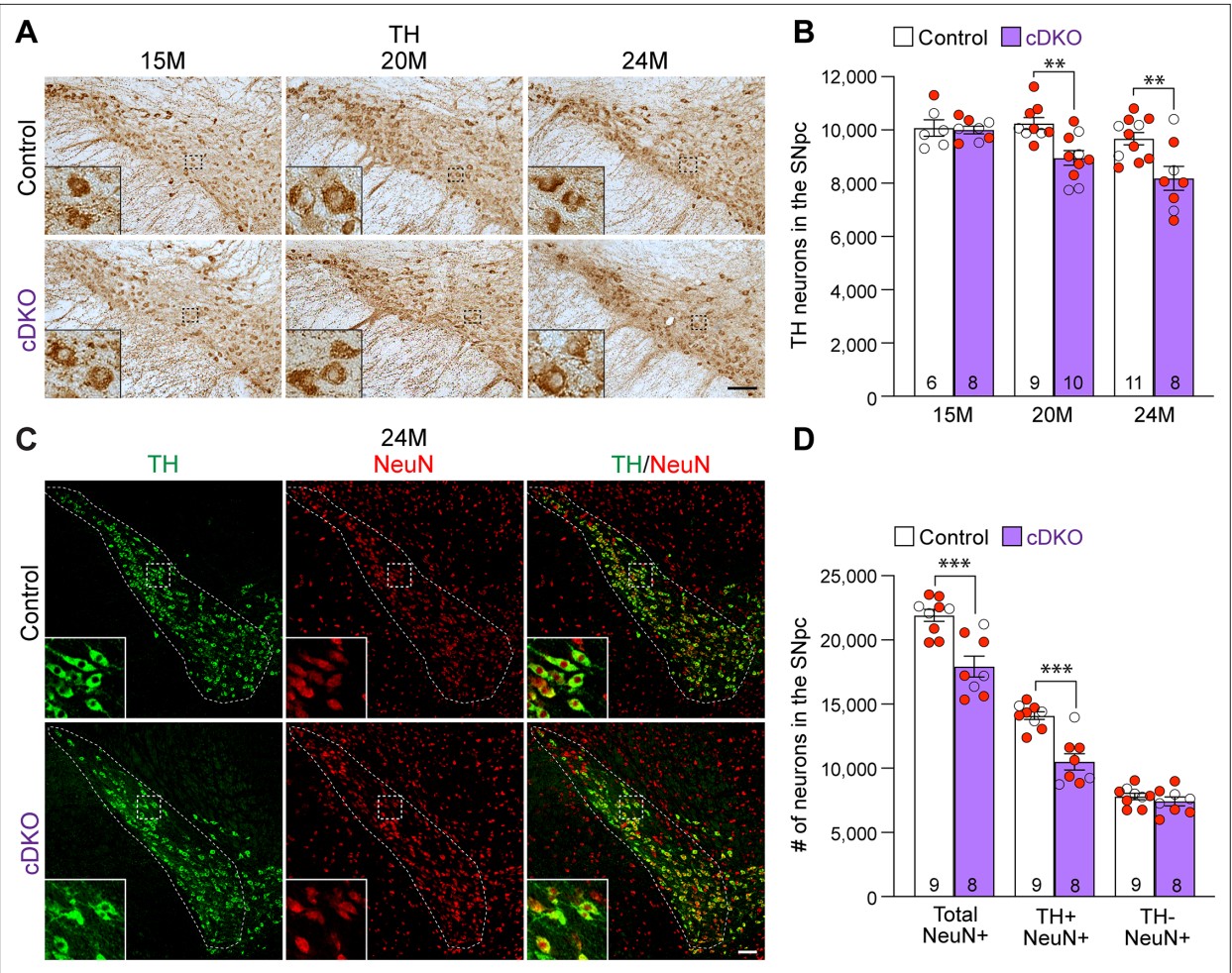

**Figure 3.** Age-dependent loss of dopaminergic (DA) neurons in the substantia nigra pars compacta (SNpc) of *Lrrk* conditional double knockout (cDKO) mice. (**A**) TH immunostaining shows TH+ DA neurons in the SNpc of *Lrrk* cDKO and littermate controls at the age of 15, 20, and 24 months. Higher power views of the boxed areas show grossly normal DA neuron morphology in *Lrrk* cDKO mice. (**B**) Quantification of TH+ DA neurons in the SNpc reveals similar numbers of DA neurons in *Lrrk* cDKO mice (10,000 ± 141) and littermate controls (10,077 ± 310, p>0.9999) at 15 months of age. At 20 months of age, the number of DA neurons in the SNpc of *Lrrk* cDKO mice (8948 ± 273) is significantly reduced compared to control mice (10,244 ± 220, $F_{1,46}$ = 16.59, p=0.0002; p=0.0041, two-way ANOVA with Bonferroni's post hoc multiple comparisons). By 24 months of age, the reduction of DA neurons in the SNpc of *Lrrk* cDKO mice (8188 ± 452) relative to controls (9675 ± 232, p=0.0010) is greater compared to 20 months of age. Raw quantification data are included in *Figure 3—source data 1*. (**C**) Immunohistological analysis of TH and NeuN shows TH+ DA neurons (green) and NeuN+ neurons (red) in the SNpc of *Lrrk* cDKO mice and controls at 24 months of age. (**D**) Quantification of NeuN+ cells in the SNpc shows that the number of NeuN+ neurons in *Lrrk* cDKO mice (17,923 ± 813) is significantly lower than that in control mice (21,907 ± 469, p=0.0006, Student's *t*-test), indicating loss of neurons in the SNpc of *Lrrk* cDKO mice. All TH+ cells are NeuN+. The number of TH+/NeuN+ cells in the SNpc of *Lrrk* cDKO mice (10,500 ± 644) is also lower compared to control mice (14,102 ± 310, p=0.0001). There is no significant difference in the number of NeuN+/TH- neurons between littermate controls (7804 ± 249) and cDKO mice (7423 ± 344, p=0.3747). The number in the column indicates the number of mice used in the study. Red-filled and open circles represent data obtained from individual male and female mice, respectively. All data are expressed as mean ± SEM. **p<0.01, ***p<0.001. Scale bar: 100 μm.

The online version of this article includes the following source data and figure supplement(s) for figure 3:

**Source data 1.** Raw quantification data of TH+ dopaminergic (DA) neurons in the substantia nigra pars compacta (SNpc) of *Lrrk* conditional double knockout (cDKO) and control mice.

**Figure supplement 1.** Independent validation of age-dependent reduction of dopaminergic (DA) neurons in the substantia nigra pars compacta (SNpc) of *Lrrk* conditional double knockout (cDKO) mice.

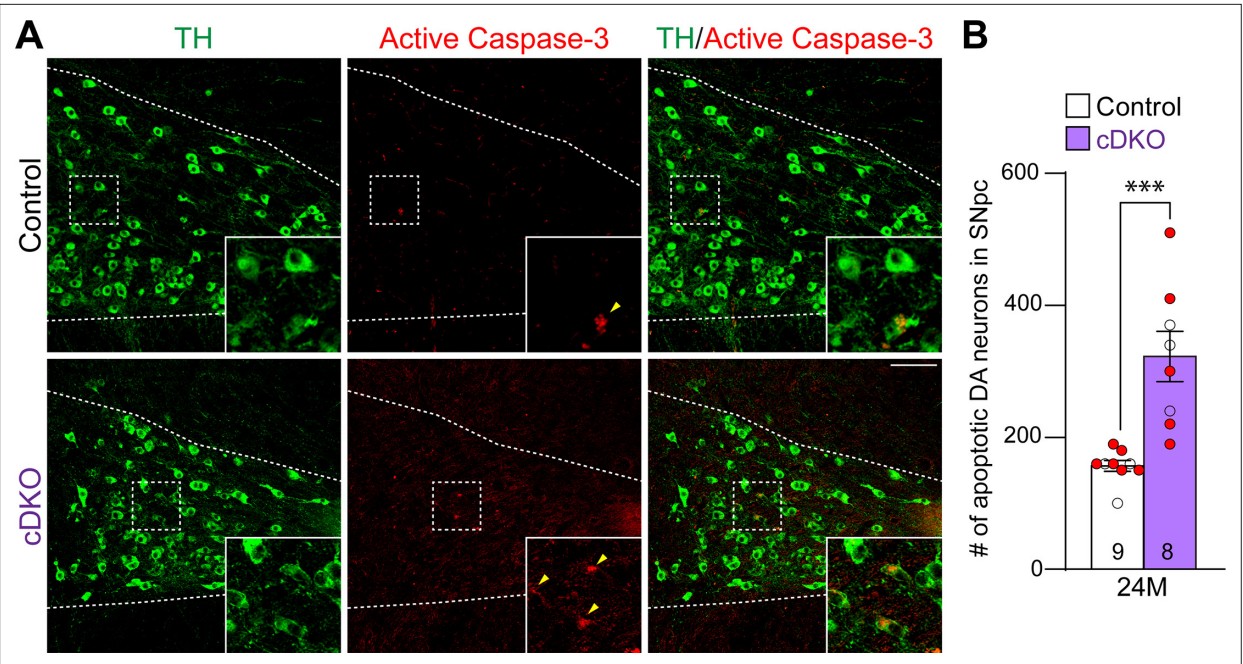

**Figure 4.** Increases in apoptotic dopaminergic (DA) neurons in the substantia nigra pars compacta (SNpc) of *Lrrk* conditional double knockout (cDKO) mice. (**A**) Representative images of TH and active Caspase-3 immunostaining show TH+ DA neurons (green) and active Caspase-3+ apoptotic cells (red) in the SNpc of *Lrrk* cDKO and control mice at the age of 24 months. (**B**) Quantification of active Caspase-3+/TH+ cells shows significant increases in apoptotic DA neurons in the SNpc of *Lrrk* cDKO mice (323 ± 38) at 24 months of age, relative to controls (157 ± 8, p=0.0004, Student's *t*-test). Raw quantification data are included in **Figure 4—source data 1**. The number in the column indicates the number of mice used in the study. Red-filled and open circles represent data obtained from individual male and female mice, respectively. All data are expressed as mean ± SEM. ***p<0.001. Scale bar: 100 μm.

The online version of this article includes the following source data for figure 4:

**Source data 1.** Raw quantification data of Caspase-3+/TH+ apoptotic dopaminergic (DA) neurons in the substantia nigra pars compacta (SNpc) of *Lrrk* conditional double knockout (cDKO) and control mice.

of *Lrrk* cDKO and littermate control mice at the ages of 15 and 24 months (**Figure 5A**). Quantitative analysis showed normal levels of TH immunoreactivity in the striatum of cDKO mice at 15 months of age but reduced levels of TH immunoreactivity in the striatum of cDKO mice at 24 moths of age (–19%, p=0.0215, Student's *t*-test), suggesting an age-dependent loss of TH+ dopaminergic terminals in the striatum (**Figure 5B**).

## Unaffected TH+ noradrenergic neurons in the LC of *Lrrk* cDKO mice

Previously, we reported the reduction of noradrenergic neurons in the locus coeruleus (LC) of *Lrrk* DKO mice (**Giaime et al., 2017**). We therefore quantified TH+ noradrenergic neurons in the LC of *Lrrk* cDKO mice at the age of 24 months. The number of TH+ cells in the LC is similar between cDKO and littermate controls (control: 3418 ± 86, cDKO: 3350 ± 99, p=0.6110, Student's *t*-test; **Figure 6A and B**). Further examination of the GFP reporter line crossed with *Slc6a3*^Cre/+ showed the lack of Cre-mediated recombination in the LC (**Figure 6C**), supporting the cell-intrinsic nature of DA neuron loss in the SNpc of aged *Lrrk* cDKO mice.

## Quantitative EM analysis of the SNpc in *Lrrk* cDKO mice

We previously reported age-dependent increases in electron-dense autophagic and autolysosomal vacuoles as well as the presence of large lipofuscin granules in the surviving SNpc neurons of *Lrrk* DKO mice beginning at 10 moths of age (**Giaime et al., 2017**; **Huang et al., 2022**). To determine whether selective inactivation of LRRK in DA neurons similarly results in the accumulation of electron-dense vacuoles in the SNpc, we performed EM analysis in the SNpc of *Lrrk* cDKO mice and littermate controls at the age of 25 months (**Figure 7**). We observed various electron-dense double membrane autophagosomes and single-membrane autolysosomes as well as lipofuscin granules composed of

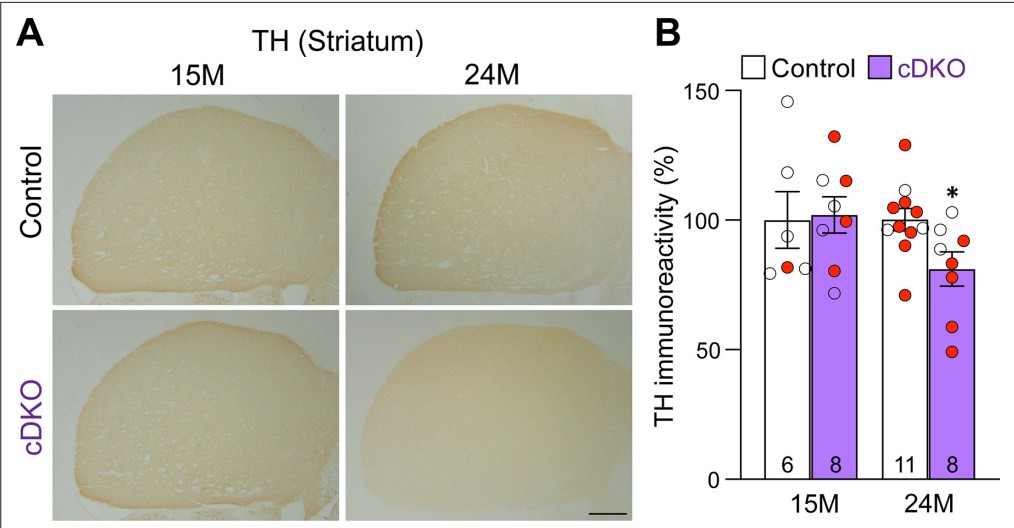

**Figure 5.** Age-dependent loss of TH+ dopaminergic (DA) terminals in the striatum of *Lrrk* conditional double knockout (cDKO) mice. (**A**) Representative TH immunostaining images in the striatum of *Lrrk* cDKO mice and littermate controls at the ages of 15 and 24 months. (**B**) Quantification of TH immunoreactivity in the striatum of *Lrrk* cDKO and control mice shows similar TH immunoreactivity at the age of 15 months (p=0.8766, Student's *t*-test), but there is a significant decrease in TH immunoreactivity in the striatum of *Lrrk* cDKO mice at 24 months of age (–19%, p=0.0215) compared to controls. The number in the column indicates the number of mice used in the study. Red-filled and open circles represent data obtained from individual male and female mice, respectively. All data are expressed as mean ± SEM. *p<0.05. Scale bar: 1 mm.

lipid-containing residues of lysosomal digestion in the SNpc of *Lrrk* cDKO and littermate control mice (*Figure 7C–F*). Interestingly, the number of electron-dense vacuoles in the SNpc is similar between *Lrrk* cDKO mice and littermate controls at the age of 25 months (control: 6.72 ± 0.43, cDKO: 6.99 ± 0.52, p=0.6839, Student's *t*-test; *Figure 7G*). We also found no significant difference in the area of electron-dense vacuoles in the SNpc between *Lrrk* cDKO and littermate controls (control: 4.43 ± 0.44 µm$^2$; cDKO: 4.60 ± 0.49 µm$^2$, p=0.8048; *Figure 7G*). The difference in accumulation of electron-dense vacuoles in the SNpc between germline DKO mice and DA neuron-restricted cDKO suggests that LRRK in non-DA neurons, possibly microglia, may play a more prominent role in the regulation of the autophagy-lysosomal pathway.

## Enhanced microgliosis in the SNpc of *Lrrk* cDKO mice

To determine whether selective inactivation of LRRK in DA neurons results in elevated microgliosis in the SNpc of *Lrrk* cDKO mice, we performed immunohistochemical analysis of Iba1, which labels microglia, and TH, which marks DA neurons and processes, thus showing the boundary of the SNpc (*Figure 8A–C*). We found that the number of Iba1+ microglia is significantly increased in the SNpc of *Lrrk* cDKO mice at 15 months of age (2541 ± 193), compared to controls (1737 ± 83; $F_{1,45}$ = 102.6, p<0.0001, two-way ANOVA with Bonferroni's post hoc multiple comparisons, p=0.0017; *Figure 8A and D*). The number of Iba1+ microglia in the SNpc of *Lrrk* cDKO is further increased compared to controls at the age of 20 (control: 2426 ± 68, cDKO: 3639 ± 127, p<0.0001; *Figure 8B and D*) and 25 months (control: 2640 ± 187, cDKO: 4089 ± 100, p<0.0001; *Figure 8C and D*). These results show that despite the selective inactivation of LRRK in DA neurons of *Lrrk* cDKO mice, microgliosis accompanies DA neuronal loss in the SNpc.

## Impaired motor coordination of *Lrrk* cDKO mice

To determine whether *Lrrk* cDKO mice show motor deficits, we performed behavioral analysis of *Lrrk* cDKO and littermate controls at the ages of 10 and 22 months using two versions of the elevated beam walk test with varying width of the beam (*Figure 9*). *Lrrk* cDKO mice at 10 months of age displayed significantly more hindlimb slips/errors (4.4 ± 0.5) and longer traversal time (7.3 ± 0.3) in the 10 mm beam walk test, relative to littermate controls, which exhibited fewer slips (2.0 ± 0.3, p=0.0005,

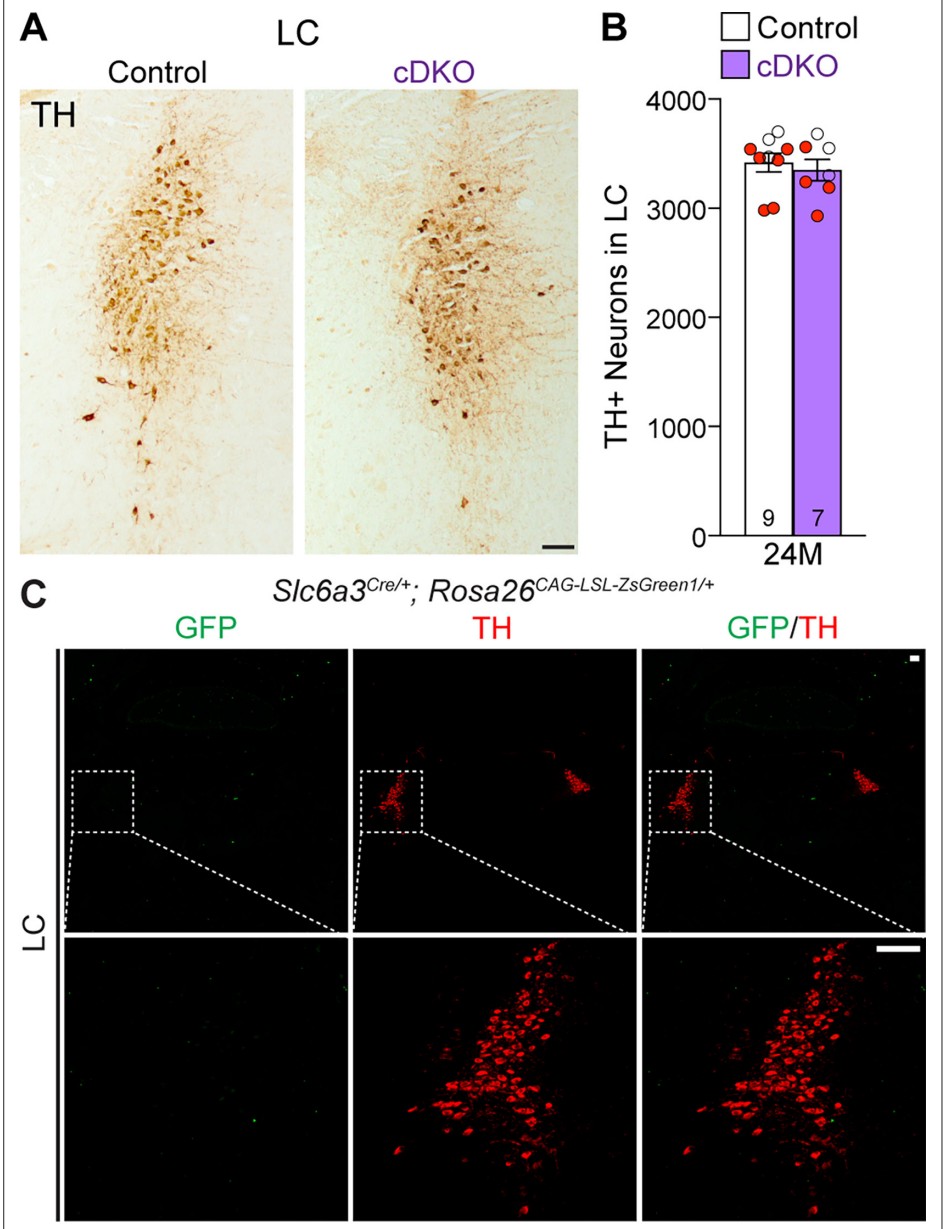

**Figure 6.** Normal number of TH+ noradrenergic neurons in the locus coeruleus (LC) of *Lrrk* conditional double knockout (cDKO) mice. (**A**) Representative images of TH+ noradrenergic neurons in the LC of *Lrrk* cDKO mice and littermate controls at 24 months of age. (**B**) Quantification of TH+ cells shows similar number of TH+ noradrenergic neurons in the LC of *Lrrk* cDKO mice (3350 ± 99) and controls (3418 ± 86, p=0.6110, Student's *t*-test). (**C**) Top: immunostaining of TH and GFP in the LC of *Slc6a3^{Cre/+}; Rosa26^{CAG-LSL-ZsGreen1/+}* mice at 2 months of age. There is no GFP+ (green) cell in the LC, indicating that *Slc6a3-Cre* is not expressed in the LC. Bottom: higher power views of the boxed areas. The number in the column indicates the number of mice used in the study. Red-filled and open circles represent data obtained from individual male and female mice, respectively. All data are expressed as mean ± SEM. Scale bar: 100 µm.

Student's *t*-test) and shorter traversal time (5.8 ± 0.4, p=0.0075; *Figure 9A*). In the 20 mm beam walk, which is less challenging than the narrower beam walk, both *Lrrk* cDKO (1.5 ± 0.2) and littermate controls (1.0 ± 0.2, p=0.0733) at 10 months of age performed well with few hindlimb slips and with no difference between *Lrrk* cDKO and control mice. In addition, the traversal time is similar between *Lrrk* cDKO (5.2 ± 0.3) and control littermates (5.1 ± 0.4, p=0.9796; *Figure 9A*). These results show that *Lrrk* cDKO mice at 10 months of age already exhibit deficits in motor coordination. However, in the

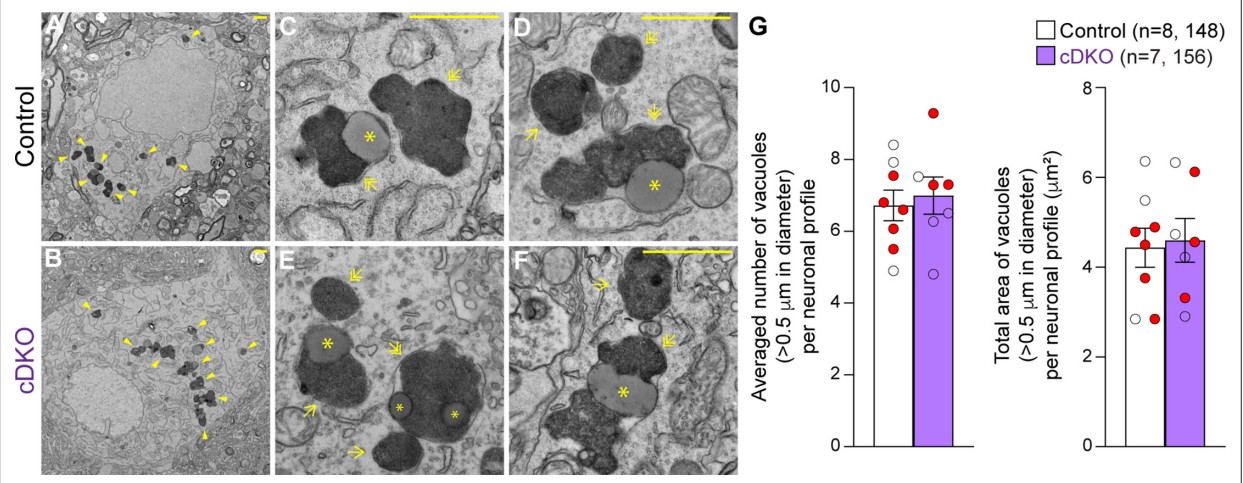

**Figure 7.** Unchanged number of electron-dense vacuoles in the substantia nigra pars compacta (SNpc) of *Lrrk* conditional double knockout (cDKO) mice. (**A, B**) Representative electron microscopy (EM) images showing electron-dense vacuoles (arrowheads) in SNpc neurons of cDKO mice and littermate controls at the age of 25 months. (**C–F**) Higher power views showing various electron-dense vacuoles, autolysosomes (single arrows), autophagosomes (double arrows), and lipid-containing vacuoles (asterisks) in SNpc neurons of littermate control (**C, D**) and cDKO (**E, F**) mice. (**G**) Left: the average number of electron-dense vacuoles (>0.5 µm in diameter) in the SNpc neuronal profiles per mouse is not significantly different between *Lrrk* cDKO mice and littermate controls at the age of 25 months (control: 6.72 ± 0.43; cDKO: 6.99 ± 0.52, p=0.6839, Student's *t*-test). Right: the total area of electron-dense vacuoles (>0.5 µm in diameter) in the SNpc neuronal profiles per mouse is similar between *Lrrk* cDKO and littermate controls (control: 4.43 ± 0.44 µm²; cDKO: 4.60 ± 0.49 µm², p=0.8048). The value in parentheses indicates the number of mice (left) and neuron profiles (right) used in the quantification. Red-filled and open circles represent data obtained from individual male and female mice, respectively. All data are expressed as mean ± SEM. Scale bar: 1 µm.

pole test *Lrrk* cDKO and control mice showed similar turning time (control: 1.2 ± 0.1; cDKO: 1.5 ± 0.1; p=0.1219) and descending time (control: 4.7 ± 0.1; cDKO: 4.7 ± 0.2; p=0.8620; *Figure 9B*). The *Lrrk* cDKO and control mice at 10 months of age exhibit similar body weight (*Figure 9C*).

We then performed the beam walk test in another group of naïve *Lrrk* cDKO and control mice at 22 months of age (*Figure 9D*). The aged *Lrrk* cDKO and control mice performed similarly in hindlimb errors (p=0.7022) and traversal time (p=0.8139) in 10 mm beam walk test. The hindlimb slips (control: 12.0 ± 3.0; cDKO: 13.8 ± 3.3) and traversal time (control: 15.3 ± 2.0; cDKO: 16.2 ± 2.8) of both genotypic groups were much higher compared to the respective groups of younger mice (p<0.0001; *Figure 9D*). Moreover, higher percentage of the aged mice (2–3 out of 8–9 per genotypic group) failed the test compared to the younger mice (0–2 mice out of 18–20 per genotypic group, *Figure 9A and D*). These results show that aged mice of both genotypes perform poorly in the challenging narrow beam walk test, suggesting that it is not optimal to reveal subtle differences in motor coordination between the genotypic groups at this age. The performance of *Lrrk* cDKO and control mice at 22 months of age in the 20 mm beam walk test was also not significantly different with similar hindlimb errors (p=0.7142) and traversal time (p=0.2223; *Figure 9D*). In the pole test, *Lrrk* cDKO mice and littermate controls also displayed similar turning time (p=0.1184) and descending time to their home cage (p=0.1413; *Figure 9E*) as well as body weight (*Figure 9F*).

## Discussion

LRRK2 mutations are linked to familial PD and are also associated with sporadic PD with the G2019S mutation being the most common but exhibiting lower penetrance and higher age of onset (e.g., 74% at age 79) (*Healy et al., 2008*). Despite the importance of LRRK2 in PD pathogenesis, whether LRRK2 mutations cause DA neurodegeneration via a loss- or gain-of-function mechanism remains unresolved, even though the distinction between these two pathogenic mechanisms is critical for directing LRRK2-based PD therapeutic development. Neither *Lrrk2*-deficient mice nor *Lrrk2* R1441C and G2019S KI mice develop dopaminergic neurodegeneration (*Tong et al., 2010*; *Tong et al., 2009*; *Yue et al., 2015*). It was proposed that the lack of brain phenotypes in *Lrrk2*-null mice might be due to the presence of LRRK1 (*Tong et al., 2010*; *Tong et al., 2012*), which is broadly expressed in the

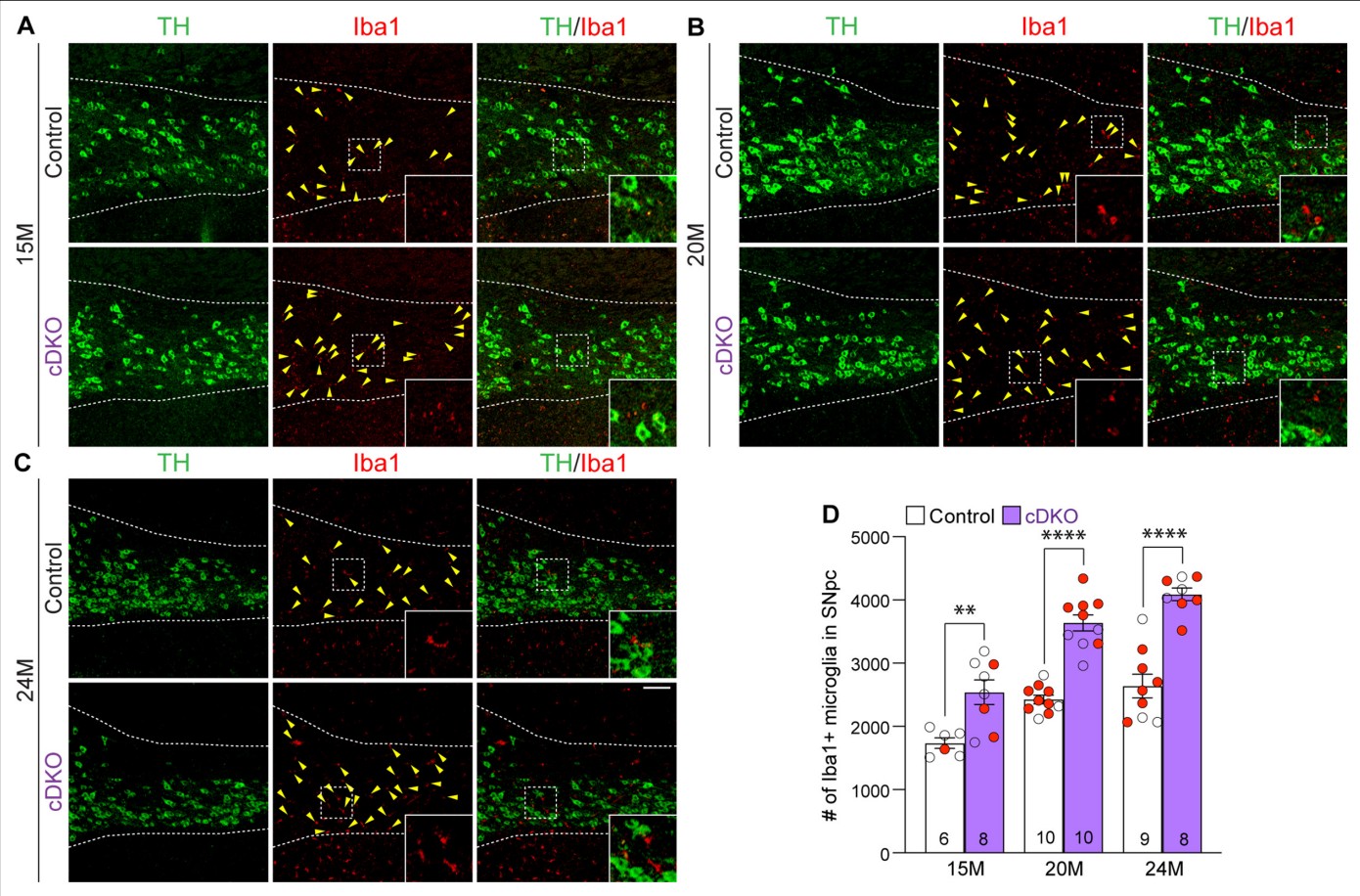

**Figure 8.** Elevated microgliosis in the substantia nigra pars compacta (SNpc) of *Lrrk* conditional double knockout (cDKO) mice. (**A–C**) Representative images of Iba1+ microglia (red, marked by yellow arrowheads) and TH+ dopaminergic neurons (green) in the SNpc of *Lrrk* DKO mice and controls at 15 (**A**), 20 (**B**), and 24 (**C**) months of age. (**D**) Quantification of Iba1+ microglia shows significant increases in the number of Iba1+ microglia in the SNpc of *Lrrk* cDKO mice compared to control mice at the ages of 15 months (control: 1737 ± 83, cDKO: 2541 ± 193; $F_{1,45}$ = 102.6, p<0.0001, p=0.0017, two-way ANOVA with Bonferroni's post hoc multiple comparisons), 20 months (control: 2426 ± 68, cDKO: 3639 ± 127, p<0.0001), and 24 months (control: 2640 ± 187, cDKO: 4089 ± 100, p<0.0001). Raw quantification data are included in *Figure 8—source data 1*. The number in the column indicates the number of mice used in the study. Red-filled and open circles represent data obtained from individual male and female mice, respectively. All data are expressed as mean ± SEM. **p<0.01, ****p<0.0001. Scale bar: 100 μm.

The online version of this article includes the following source data for figure 8:

**Source data 1.** Raw quantification data of Iba1+ microglia in the substantia nigra pars compacta (SNpc) of *Lrrk* conditional double knockout (cDKO) and control mice.

brain including the midbrain (https://www.proteinatlas.org/ENSG00000154237-LRRK1/brain). Indeed, germline deletions of *Lrrk2* and *Lrrk1* result in an age-dependent loss of DA neurons and increases in apoptosis and microgliosis in the SNpc of *Lrrk* DKO mice (*Giaime et al., 2017*; *Huang et al., 2022*). However, the pleiotropic roles of LRRK1 and LRRK2 in the kidney (*Tong et al., 2010*; *Tong et al., 2012*; *Tong and Shen, 2012*), lung (*Tian et al., 2021*), and bone (*Xing et al., 2013*; *Guo et al., 2017*), and the observed earlier mortality and lower body weight in *Lrrk* DKO mice raised the possibility that DA neurodegeneration in *Lrrk* DKO mice may be due to poor health.

To address this question, we generated floxed *Lrrk1* and *Lrrk2* mice (*Figure 1Figure 1—figure supplements 1–8*) and DA neuron-specific *Lrrk* cDKO mice using the *Slc6a3-Cre* KI allele (*Bäckman et al., 2006*) to delete floxed *Lrrk1* and *Lrrk2* regions selectively in DA neurons (*Figure 2*). Using a GFP reporter line, we found that *Slc6a3-Cre* drives Cre-mediated recombination in almost all DA neurons in the SNpc. Unlike germline DKO mice (*Giaime et al., 2017*), DA neuron-specific *Lrrk* cDKO mice exhibit normal body weight and mortality during mouse lifespan. Importantly, *Lrrk* cDKO mice develop an age-dependent, progressive DA neurodegeneration, as evidenced by the normal number

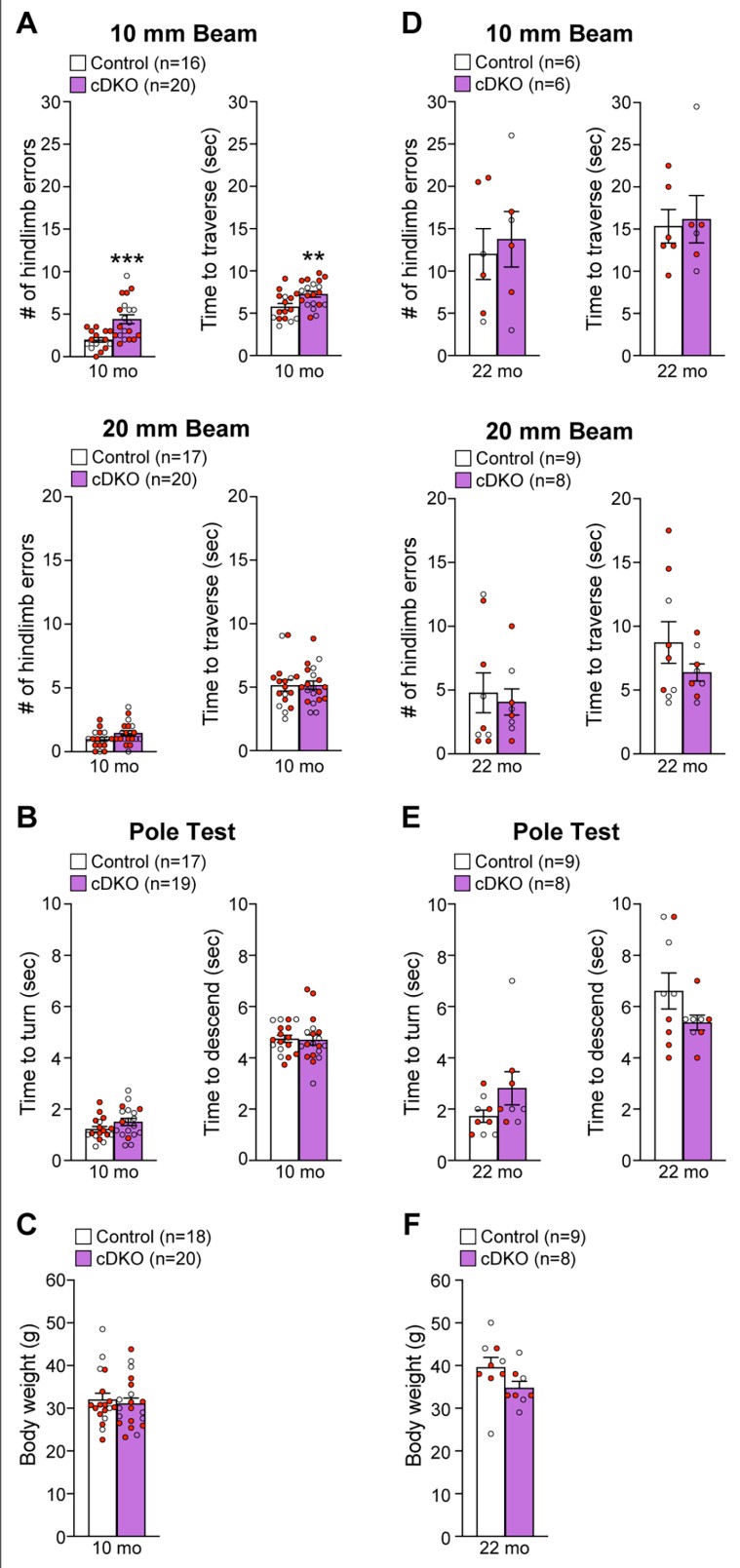

**Figure 9.** Impairment of motor coordination in *Lrrk* conditional double knockout (cDKO) mice. (**A**) In the 10 mm beam walk test, compared to control mice, *Lrrk* cDKO mice at 10 months of age exhibit markedly more hindlimb slips (control: 2.0 ± 0.3; cDKO: 4.4 ± 0.5; p=0.0005, Student's *t*-test) and longer traversal time (control: 5.8 ± 0.4; cDKO: 7.3 ± 0.3; p=0.0075). In the less challenging 20 mm beam walk test, there is no significant difference in

*Figure 9 continued on next page*

*Figure 9 continued*

the number of hindlimb slips (control: 1.0 ± 0.2; cDKO: 1.5 ± 0.2; p=0.0733) and traversal time (control: 5.1 ± 0.4; cDKO: 5.2 ± 0.3; p=0.9796) between *Lrrk* cDKO and control mice. (**B**) In the pole test, *Lrrk* cDKO and control mice at 10 months of age display similar turning time (control: 1.2 ± 0.1; cDKO: 1.5 ± 0.1; p=0.1219) and descending time (control: 4.7 ± 0.1; cDKO: 4.7 ± 0.2; p=0.8620). (**C**) *Lrrk* cDKO (31.1 ± 1.3) and control (32.0 ± 1.5; p=0.6410) mice at 10 months of age show similar body weight. (**D**) *Lrrk* cDKO mice and control mice at 22 months of age in the 10 mm beam walk test show similar hindlimb slips (control: 12.0 ± 3.0; cDKO: 13.8 ± 3.3; p=0.7022) and traversal time (control: 15.3 ± 2.0; cDKO: 16.2 ± 2.8; p=0.8139). In the 20 mm team walk test, there is also no difference in hindlimb slips (control: 4.8 ± 1.6; cDKO: 4.1 ± 1.0; p=0.7142) and traversal time (control: 8.7 ± 1.6; cDKO: 6.4 ± 0.7; p=0.2223) between *Lrrk* cDKO and control mice. (**E**) In the pole test, *Lrrk* cDKO mice at 22 months of age exhibit similar turning time (control: 1.7 ± 0.2; cDKO: 2.8 ± 0.7; p=0.1184) and descending time (control: 6.6 ± 0.7; cDKO: 5.4 ± 0.3; p=0.1413) compared to control mice. (**F**) *Lrrk* cDKO mice (34.8 ± 1.5) have similar body weight as control mice (39.6 ± 2.4; p=0.1194). The number in parentheses indicates the number of mice used in the study. Red-filled and open circles represent data obtained from individual male and female mice, respectively. All data are expressed as mean ± SEM. **p<0.01, ***p<0.001. Raw behavior data are included in *Figure 9—source data 1*.

The online version of this article includes the following source data for figure 9:

**Source data 1.** Raw quantification data of the beam walk and the pole tests by *Lrrk* conditional double knockout (cDKO) and control mice.

---

of DA neurons at the age of 15 months and the progressive reduction of DA neurons in the SNpc at the ages of 20 and 24 months (*Figure 3*), accompanied with increases in apoptotic DA neurons in the SNpc (*Figure 4*) and decreases in DA terminals in the striatum (*Figure 5*), whereas the number of noradrenergic neurons in the LC is unchanged, consistent with the lack of Cre-mediated recombination in the LC by *Slc6a3-Cre* (*Figure 6*). Moreover, *Lrrk* cDKO mice exhibit impaired performance in a narrow beam walk, a behavioral paradigm sensitive to altered motor coordination (*Figure 9*). Furthermore, microgliosis is elevated in the SNpc of *Lrrk* cDKO mice, even though LRRK expression in microglia is unaffected in these mice, suggesting a cell-extrinsic microglial response to pathophysiological changes in DA neurons (*Figure 8*). The molecular mechanism by which LRRK supports cell-autonomous DA neuron survival is unknown. Single-cell RNA sequencing of dissected ventral midbrains may permit identification of genes and pathways that are selectively altered in DA neurons of *Lrrk* cDKO mice before and after the onset of DA neurodegeneration.

Interestingly, quantitative EM analysis showed similar numbers of electron-dense vacuoles in the SNpc of *Lrrk* cDKO and control mice at 25 months of age (*Figure 7*), in contrast to age-dependent increases in vacuoles in the SNpc of *Lrrk* DKO mice (*Giaime et al., 2017*; *Huang et al., 2022*), suggesting non-DA neurons (e.g., microglia), in which LRRK expression is unaffected in DA neuron-specific *Lrrk* cDKO mice but is absent in germline DKO mice, may contribute to this phenotypic difference. Development of microglia- and/or astrocyte-specific *Lrrk* cDKO mice would permit further dissection of cell-autonomous and non-cell-autonomous roles of LRRK in DA neuron survival and determine whether glial LRRK plays a more important role in the regulation of the autophagy-lysosomal pathway and protein turnover. It would also be interesting to test whether reducing LRRK expression or function affects accumulation and aggregation of human mutant α-synuclein and tau in the aging brain, as both Lewy bodies and tauopathies are associated with PD patients carrying LRRK2 mutations (*Zimprich et al., 2004*).

The toxic gain-of-function pathogenic mechanism was initially proposed based on in vitro kinase assays using recombinant mutant LRRK2, which showed that R1441C and G2019S resulted in increases in autophosphorylation and phosphorylation of a generic substrate (*West et al., 2005*). Subsequent biochemical studies further reported that LRRK2 mutations (e.g., R1441C/G, G2019S) enhanced kinase activity, leading to elevated levels of pSer1292-LRRK2 (*Sheng et al., 2012*) as well as pT73-Rab10 and pS106-Rab12 (*Steger et al., 2016*). While these biochemical changes are excellent biomarkers of LRRK2 mutations, there is no experimental evidence showing that increased phosphorylation of LRRK2 substrates drives DA neurodegeneration.

The loss-of-function pathogenic mechanism was prompted by mouse knockout findings showing that germline inactivation of the *Lrrk* genes results in age-dependent loss of DA neurons in the SNpc and DA terminals in the striatum (*Giaime et al., 2017*; *Huang et al., 2022*). The current study demonstrated a cell-intrinsic role of LRRK in support of DA neuron survival in the SNpc of the aging brain,

providing an unequivocal genetic evidence of an essential, cell-autonomous requirement of LRRK in DA neurons. While most transgenic mice overexpressing mutant LRRK2 did not produce neurodegeneration or DA neuron loss (*Li et al., 2009*; *Lin et al., 2009*; *Li et al., 2010*; *Melrose et al., 2010*; *Daher et al., 2012*; *Tsika et al., 2014*; *Liu et al., 2015*; *Mikhail et al., 2015*; *Xiong et al., 2017*), two studies reported that overexpression of LRRK2 G2019S but not R1441C resulted in DA neurodegeneration (*Ramonet et al., 2011*; *Xiong et al., 2018*). These transgenic mouse results are intriguing as R1441C is a more potent causative mutation (highly penetrant, in a critical amino acid residue that harbors three distinct PD mutations), whereas G2019S is known to be a weaker mutation with a lower penetrance and an older age of disease onset.

It remains possible that the relevant physiological role of LRRK in the protection of DA neuron survival during aging may be irrelevant to how LRRK2 mutations cause DA neurodegeneration and PD. Two reports of genomic database analysis suggested that loss-of-function *Lrrk2* and *Lrrk1* variants are not associated with PD (*Whiffin et al., 2020*; *Blauwendraat et al., 2018*), which were sometimes taken as conclusive evidence for LRRK2 mutations not being loss of function. One study was a predictive analysis using several available sequencing databases (141,456 individuals sequenced in the Genome Aggregation Database, 49,960 exome-sequenced individuals from the UK Biobank, and more than 4 million participants in the 23andMe genotyped dataset), and the authors cautiously stated that heterozygous predicted loss-of-function variants are not strongly associated with PD (*Whiffin et al., 2020*). Another study analyzed next-generation sequencing data from 11,095 PD patients and 12,615 controls, and found that *Lrrk1* loss-of-function variants were identified in 0.205% PD cases and 0.139% of controls, whereas *Lrrk2* loss-of-function variants were found in 0.117% of PD cases and 0.087% of controls (*Blauwendraat et al., 2018*). In contrast to linkage studies of large pedigrees with mutations segregating completely with the disease but being absent among control populations, these genomic database analysis studies are considerably weaker with the results suggestive but highly inconclusive. For example, while the occurrence of these loss-of-function variants among PD patients is not statistically different from those among control populations, a larger sample size may change this outcome. Furthermore, such sequencing data tend to be from highly heterogeneous populations, contributing to further complexity. Moreover, it is quite possible that haploinsufficiency or heterozygous complete loss-of-function mutations are not sufficient to cause the disease; rather a dominant negative mechanism, coupled with loss of function missense mutations, may be at play, reducing overall protein function further, especially if dimers are the functional entities. Therefore, these studies are interesting but inconclusive as to whether loss-of-function mutations in LRRK2 or LRRK1 cause or do not cause PD.

Genetically, it is well known that missense mutations may gain a toxic function or may cause a partial loss of function in *cis* and a dominant negative inhibition of wild-type protein in *trans*, further reducing its function. Missense mutations in the *PSEN* genes linked to familial Alzheimer's disease are good examples of such a loss-of-function coupled with a dominant negative mechanism (*Shen and Kelleher, 2007*; *Heilig et al., 2010*; *Heilig et al., 2013*; *Zhou et al., 2017*; *Watanabe and Shen, 2017*; *Sun et al., 2017*; *Kelleher and Shen, 2017*). While PD-linked *Lrrk2* mutations are mostly dominantly inherited missense mutations, though homozygous R1441H carriers have been reported (*Takanashi et al., 2018*), variants in LRRK1 have also been reported with inconclusive pathogenicity (*Schulte et al., 2014*; *Haugarvoll et al., 2007*). The notion that mutant LRRK2 may act in a dominant negative manner to inhibit the activity of wild-type LRRK2 is supported by structural and functional studies of LRRK2 as homodimers or heterodimers with LRRK1 (*Deng et al., 2008*; *Myasnikov et al., 2021*; *Zhang et al., 2019*; *Greggio et al., 2008*; *Sen et al., 2009*; *Jorgensen et al., 2009*; *Dachsel et al., 2010*). Furthermore, LRRK2 G2385R variant, a risk factor of PD (*Farrer et al., 2007*; *Kim et al., 2010*; *An et al., 2008*; *Xie et al., 2014*), has been reported as a partial loss-of-function mutation (*Rudenko et al., 2012*; *Carrion et al., 2017*). Future studies are needed to reconcile the biochemical findings of elevated kinase activity by LRRK2 mutations and the genetic findings of an essential physiological role of LRRK in DA neuron survival, and to

determine whether LRRK2 mutations impair this relevant physiological function in protecting DA neurons while increasing kinase activity.

# Materials and methods

## Key resources table

| Reagent type (species) or resource | Designation | Source or reference | Identifiers | Additional information |
|---|---|---|---|---|
| Gene (*Mus musculus*) | *Lrrk1* | Ensembl Genome Database: ENSMUSG00000015133 | | |
| Gene (*M. musculus*) | *Lrrk2* | Ensembl Genome Database: ENSMUSG00000036273 | | |
| Strain, strain background (*M. musculus*) | B6129SF1/J | The Jackson Laboratory | RRID:IMSR_JAX:101043 | Strain #101043 |
| Strain, strain background (*M. musculus*) | Floxed *Lrrk1* | This paper | | Maintained in B6/129 hybrid background |
| Strain, strain background (*M. musculus*) | Floxed *Lrrk2* | This paper | | Maintained in B6/129 hybrid background |
| Strain, strain background (*M. musculus*) | B6.SJL-*Slc6a3*$^{tm1.1(cre)bkmn}$/J | The Jackson Laboratory | RRID:IMSR_JAX:006660 | Strain # 006660; common name: DAT$^{IREScre}$ |
| Strain, strain background (*M. musculus*) | B6.Cg-Tg(*ACTFLPe*)9205Dym/J | The Jackson Laboratory | RRID:IMSR_JAX:005703 | Strain # 005703; common name: ACTB:FLPe B6J |
| Strain, strain background (*M. musculus*) | B6.C-Tg(*CMV-cre*)1Cgn/J | The Jackson Laboratory | RRID:IMSR_JAX:006054 | Strain # 006054; common name: CMV-Cre |
| Strain, strain background (*M. musculus*) | B6.Cg-*Gt(ROSA)26Sor*$^{tm6(CAG-ZsGreem1)Hze}$/J | The Jackson Laboratory | RRID:IMSR_JAX:007906 | Strain # 007906; common name: Ai6RCL-ZsGreen |
| Cell line (*M. musculus*) | MKV6.5 embryonic stem cells | A gift from Dr. Rudy Jaenisch's lab at MIT | | B6129SF1/J |
| Transfected construct (*M. musculus*) | pLM8 | This paper | | Details provided in **Supplementary file 1** |
| Transfected construct (*M. musculus*) | pLRRK2#8 | This paper | | Details provided in **Supplementary file 2** |
| Antibody | Anti-LRRK1 (rabbit polyclonal) | Alomone Lab | Cat# ANR-101; RRID:AB_2756700 | WB: 1:1000 |
| Antibody | Anti-LRRK2 (rabbit monoclonal) | abcam | Cat# Ab133474; RRID:AB_2713963 | WB: 1:1000 |
| Antibody | Anti-α-Vinculin (mouse monoclonal) | Millipore | Cat# 05-386; RRID:AB_309711 | WB: 1:2000 |
| Antibody | Anti-GFP (rabbit polyclonal) | abcam | Cat# ab290; RRID:AB_303395 | IF: 1:1000 |
| Antibody | Anti-TH (mouse monoclonal) | SantaCruz | Cat# Sc-25269; RRID:AB_628422 | IF: 1:50 |
| Antibody | Anti-TH (rabbit polyclonal) | abcam | Cat# Ab112; RRID:AB_297840 | IHC: 1:750 |
| Antibody | Anti-NeuN (rabbit monoclonal) | Cell Signaling Technology | Cat# 12943S; RRID:AB_2630395 | IF: 1:400 |
| Antibody | Anti-cleaved caspase-3 (rabbit monoclonal) | Cell Signaling Technology | Cat# 9661; RRID:AB_2341188 | IF: 1:150 |
| Antibody | Anti-Iba1 (rabbit polyclonal) | Wako | Cat# 019-19741; RRID:AB_839504 | IF: 1:500 |
| Antibody | Goat anti-rabbit (goat IgG, IRdye800 coupled) | LI-COR Biosciences | Cat# 926-32211; RRID:AB_2651127 | WB: 1:20,000 |
| Antibody | Goat anti-mouse (goat IgG, IRdye680 coupled) | LI-COR Biosciences | Cat# 926-68020; RRID:AB_2651128 | WB: 1:20,000 |

*Continued on next page*

*Continued*

| Reagent type (species) or resource | Designation | Source or reference | Identifiers | Additional information |
|---|---|---|---|---|
| Antibody | Goat anti-mouse (goat polyclonal, Alexa Fluor 488 conjugated) | Thermo Fisher Scientific | Cat# A-11001; RRID:AB_2534069 | IF: 1:250 |
| Antibody | Goat anti-rabbit (goat polyclonal, Alexa Fluor 555 conjugated) | Thermo Fisher Scientific | Cat# A-32732; RRID:AB_2633281 | IF: 1:250 |
| Antibody | Goat anti-rabbit (goat polyclonal, Alexa Fluor 488 conjugated) | Thermo Fisher Scientific | Cat# A-11034; RRID:AB_2576217 | IF: 1:250 |
| Antibody | Goat anti-mouse (goat polyclonal, Alexa Fluor 555 conjugated) | Thermo Fisher Scientific | Cat# A-21424; RRID:AB_141780 | IF: 1:250 |
| Antibody | Goat anti-rabbit (goat IgG, Biotinylated) | Thermo Fisher Scientific | Cat# BA-1000; RRID:AB_2313606 | IHC: 1:250 |
| Recombinant DNA reagent | *pGEM-T* (vector) | Promega | Cat# A1360 | |
| Recombinant DNA reagent | *PgkneoF2L2DTA* (plasmid) | Addgene | RRID:Addgene_13445 | |
| Recombinant DNA reagent | *pCAGGS-flpE-puro* (Plasmid) | Addgene | RRID:Addgene_20733 | |
| Recombinant DNA reagent | *pBluescript II KS (+)* (vector) | Agilent | Part number: 212207 | |
| Recombinant DNA reagent | *LFNT-tk/pBS* (Plasmid) | A gift from Dr. Susumu Tonegawa's lab at MIT | | |
| Sequence-based reagent | Primers | This paper | PCR primers | See *Appendix 1—table 1* |
| Commercial assay or kit | Poly(A)Purist MAG Kit | Thermo Fisher | Cat# AM1922 | |
| Commercial assay or kit | Prime-It II random labeling Kit | Agilent | Cat# 300385 | |
| Commercial assay or kit | DAB peroxidase substrate kit | Vector Laboratories | Cat# SK-4100 | |
| Chemical compound, drug | TRI reagent | MilliporeSigma | T9424 | |
| Chemical compound, drug | Superscript III reverse transcriptase | Thermo Fisher | Cat# 18080093 | |
| Chemical compound, drug | Protease inhibitor cocktail | Sigma | Cat# P8340 | |
| Chemical compound, drug | Phosphatase inhibitor cocktail | Sigma | Cat# P0044 | |
| Chemical compound, drug | Vectastain elite ABC reagent | Vector Laboratories | Cat# SK-6100 | |
| Software, algorithm | Prism 9 | GraphPad | 9.0 | |
| Software, algorithm | CellSens Entry | Olympus | 1.5 | |
| Software, algorithm | Image-Studio | Odyssey | 5.2 | |
| Software, algorithm | ImageJ Fiji | NIH | 1.50i | |
| Other | Amersham Hybond-nylon membrane | GE Healthcare | RPN303N | Used for RNA transfer in northern (see 'Materials and methods' for the details) |
| Other | Autoradiography film, Hyperfilm | Amersham | E3018 | Used for detection of radioactive signals in northern (see 'Materials and methods for the details) |

## Mice

All animal use was approved by the IACUC committees of Harvard Medical School and Brigham and Women's Hospital (#2016N000120) and conformed to the USDA Animal Welfare Act, PHS Policy on Humane Care and Use of Laboratory Animals, the 'ILAR Guide for the Care and Use of Laboratory Animals', and other applicable laws and regulations. Mice were housed in constant humidity- and temperature-controlled rooms and maintained on a 12 hr light/dark cycle and were given standard rodent chow and water. Mice of both sexes at multiple ages, from 2 months to 25 months, were used. *Lrrk1*^F/F and the resulting germline deleted *Lrrk1*^Δ/Δ mice, and *Lrrk2*^F/F and the resulting germline deleted *Lrrk2*^Δ/Δ mice were generated and thoroughly validated at the genomic DNA, mRNA, and protein levels. *Slc6a3-Cre* (The Jackson Laboratory, IMSR_JAX:006660), *ACTB-FLPe* (IMSR_JAX:005703), *CMV-Cre* (IMSR_JAX:006054), and *Rosa26*^CAG-LSL-ZsGreen1 (IMSR_JAX:007906) mice used in the current study were previously characterized and reported (*Bäckman et al., 2006*; *Rodríguez et al., 2000*; *Schwenk et al., 1995*; *Madisen et al., 2010*). Lrrk1/Lrrk2 cDKO mice (*Lrrk1*^F/F; *Lrrk2*^F/F; *Slc6a3*^Cre/+) and littermate controls (*Lrrk1*^F/F; *Lrrk2*^F/F) were obtained from multiple breeding cages of *Lrrk1*^F/F; *Lrrk2*^F/F and cDKO mice. All mice were maintained on the C57BL6 and 129 hybrid genetic background (F1: IMSR_JAX:101043). All phenotypic analyses were performed in a genotype-blind manner, as previously described (*Kang et al., 2021*).

## Generation of targeted and floxed *Lrrk1* and *Lrrk2* alleles

The generation and validation of the *Lrrk1* and *Lrrk2* targeting vectors, the targeted, floxed, and deleted *Lrrk1* and *Lrrk2* alleles by Southern, northern, RT-PCR, and the sequences of the floxed *Lrrk1* and *Lrrk2* alleles are included in *Figure 1—figure supplements 1–9*, *Supplementary files 1 and 2*.

### *Lrrk1*

To generate the *Lrrk1* targeting vector, we first PCR amplified the *left middle* homologous region (2079 bp) containing partial intron 26, exon 27, and partial intron 27 of *Lrrk1* from mouse BAC DNA (clone RP23-213J23, BACPAC Resources Center) using primers P3 and P4. The PCR fragment was subcloned into the *pGEM-T* vector (A1360, Promega) to generate pLM1 (for details, see ). The *right middle* homologous region (3403) containing *Lrrk1* genomic region from partial intron 27 to partial intron 29 was amplified by PCR using primers P5 and P6, and then subcloned into the *pGEM-T* vector (pLM2). The *left middle* homologous region containing *pGEM-T* plasmid was then digested with *Not*I and *Kpn*I (endogenous *Kpn*I site in the intron 27) and subcloned into the *Not*I and *Kpn*I sites of the *right middle* homologous region containing *pGEM-T* plasmid (pLM3). Then, the *middle* homologous region (5.5 kb), from partial intron 26 to partial intron 29, was released by *Not*I and *Sac*II digestions and was blunted by Klenow, and then subcloned into the *Sma*I site of *PgkneoF2L2DTA* vector (#13445, Addgene) to generate the *middle* homologous region-*PgkneoF2L2DTA* plasmid (pLM4).

The *left* homologous region (2016 bp) containing partial intron 25, exon 26, and partial intron 27 of *Lrrk1* was PCR amplified from mouse BAC DNA (clone RP23-213J23, BACPAC Resources Center) using primers P1 and P2. The PCR fragment was digested with *Sac*II (introduced by P1) and *Not*I (introduced by P2 along with *Hind*III), and was subcloned into the *Sac*II and *Not*I sites of the *PgkneoF2L2DTA* vector (pLM5). The *right* homologous region (3131 bp) containing partial intron 29, exon 30, and partial intron 30 of *Lrrk1* was PCR amplified from mouse BAC DNA (clone RP23-213J23) using primers P7 and P8. The PCR fragment was digested with *Sal*I (introduced by P7) and *Hind*III (introduced by P8), and was subcloned into the *Sal*I and *Hind*III sites of the *PgkneoF2L2DTA* vector (pLM6).

The *left* homologous region in the *PgkneoF2L2DAT* vector (pLM5) was digested with *Sac*II and *Not*I, and subcloned into the *Sac*II and *Not*I sites of the *right* homologous region containing *PgkneoF2L2DTA* plasmid (pLM6) to generate pLM7. Finally, the *loxP-middle* homologous region-*Pgk-Neo-loxP* fragment was released from pLM4 by *Not*I and *Sal*I digestions, and subcloned into the *Not*I and *Sal*I sites of pLM7 to generate the final target vector (pLM8), which contains two loxP sites (intron 26 and intron 29). Upon Cre-mediated recombination, the endogenous *Lrrk1* genomic sequence flanked by the 5' loxP site (1288 bp upstream of *Lrrk1* exon 27) to the 3' loxP site (1023 bp downstream of exon 29) is excised. The *Pgk-neo* cassette flanked by two FRT sites is under the control of the mouse *phosphoglycerate kinase 1* (*Pgk*) promoter and contains the *bovine growth hormone* polyA signal. To enhance the ratio of ES cells carrying homologous recombination events instead of random insertion

of the targeting vector (*Yu et al., 2000*), the negative selection *Pgk-DTA* cassette, which encodes diphtheria toxin A chain, was also included in the *Lrrk1* targeting vector.

The *Lrrk1* targeting vector was linearized by *Xho*I digestion, and then electroporated into MKV6.5 ES cells (a gift from R. Jaenisch), which were derived from B6/129 F1 mice (The Jackson Laboratory, IMSR_JAX:101043). G418 was applied to the culture at 150 µg/ml 24 hr later, and after 6 d of G418 selection, the surviving ES clones (480) were picked and screened by Southern analysis using *Hind*III digestion of genomic DNA followed by hybridization with the 5' external and 3' external probes to confirm proper recombination events in the 5' and 3' homologous regions, respectively. Twenty-five ES cell clones were confirmed to carry the proper homologous recombination events at the 5' homologous region, giving rise to the 17.0 kb and the 4.8 kb bands, which represent the wild-type and the targeted allele, respectively (*Figure 1D*), and the 3' homologous region, giving rise to the 17.0 kb and the 14.1 kb bands, which represent the wild-type and the targeted allele, respectively. We then expanded the selected four ES cell clones (3A11, 3D8, 3H1, and 3H6) and further verified by Southern analysis using the 5' and 3' external, and *neo* probes. Two ES cell clones (3D8 and 3H6) were transfected with *pCAGGS-flpE-puro* (#20733, Addgene) to delete the *Pgk-neo* cassette by FLP recombinase. Three resulting ES cell clones (3D8C5, 3D8E5, and 3H6G7) were confirmed by Southern analysis using the 5' and 3' external probes and the *neo* probe to confirm the floxed *Lrrk1* allele by the deletion of the *Pgk-neo* cassette.

## Lrrk2

To generate the *Lrrk2* targeting vector, the *left* homologous region (2579 bp) containing the *Lrrk2* promoter region was PCR amplified from mouse BAC DNA (clone RP23-526A2, BACPAC Resources Center) using primers P9 and P10 and subcloned into the *pGEM-T* vector to generate pLRRK2#1 (for details, see ). The *BamHI-loxP-NheI-SpeI* fragment (56 bp), which was generated by annealing two complementary oligos, P39 and P40, was introduced to the *BamHI* and *SpeI* sites of pLRRK2#1 to generate pLRRK#2. The *middle* homologous region, containing genomic sequences from the promoter region to partial intron 2, was PCR amplified from mouse BAC DNA (clone RP23-526A2) using primers P11 and P12, digested with *Xba*I and *Not*I, and subcloned into the *Xba*I and *Not*I sites of the *pGEM-T* vector to generate pLRRK2#3, which was digested with *Xba*I and *Not*I, and subcloned into the *Xba*I and *Not*I sites of pLRRK2#2 to generate pLRRK2#4.

The *right* homologous region (3503 bp), which contains partial intron 1, exon 2, and partial intron 2, was PCR amplified from mouse BAC DNA (clone RP23-526A2) using primers P13 and P14, and the PCR fragment was subcloned into the *EcoR*V site of *pBluescript II KS* (+) vector (212207, Agilent) to generate pLRRK2#5. The *right* homologous region in pLRRK2#5 was digested with *BamH*I followed by Klenow, and then digested with *Cla*I. The released fragment was subcloned into the *EcoR*V and *Cla*I sites of the *PgkneoF2L2DTA* vector to generate pLRRK2#6. *LFNT-tk/pBS* plasmid (a gift from S. Tonegawa) was digested with *Sac*II (followed by Klenow) and *Not*I (followed by *Ssp*I digestion to make it easier to isolate the *Not*I-*Sac*II/KN fragment) to release the *loxP-FRT-Pgk-neo-loxP-FRT* fragment (2928 bp), and then subcloned into the *Xba*I (followed by Klenow) and *Not*I sites of pLRRK2#6 to generate pLRRK2#7. Finally, pLRRK2#4 containing the *left-loxP-middle* homologous regions was digested with *EcoR*V and *Not*I, and subcloned into *Ale*I and *Not*I sites of pLRRK2#7 containing the *loxP-FRT -Pgk-neo- loxP-FRT -right* homologous region to generate the final targeting vector, which contains the loxP sites in the promoter region (1768 bp upstream of the transcription initiation site) and in *Lrrk2* intron 2 (878 bp downstream of exon 2). Upon Cre-mediated recombination, the floxed endogenous *Lrrk2* genomic sequences from the 5' loxP site (1768 bp upstream of *Lrrk2* transcription) to the 3' loxP site in intron 2 (878 bp downstream of exon 2) are excised. The negative selection *Pgk-DTA* cassette is also included in the *Lrrk2* targeting vector.

The *Lrrk2* targeting vector was linearized by *Ahd*I digestion and then electroporated into MKV6.5 ES cells. G418 was applied to the culture at 150 ug/ml 24 hr later, and after 6 d of G418 selection, the surviving ES clones (480) were picked and screened by Southern analysis using *Nhe*I digestion of genomic DNA followed by hybridization with the 5' external probe (753 bp, PCR amplified using P19 and P20). Southern analysis confirmed that two ES cell clones (N24 and N34) carry the proper recombination event in the 5' homologous arm, giving rise to 11.6 kb and 3.6 kb bands, which represent the wild-type and targeted alleles, respectively (*Figure 1F*), followed by genomic PCR confirmation for the proper recombination in the right arm.

Three (3D8C5, 3A11, 3H6) and one (N24) ES clones for *Lrrk1* and *Lrrk2*, respectively, were microinjected into C57BL/6 mouse blastocysts to generate chimera mice, which were bred with B6/129 F1 mice to produce heterozygous floxed *Lrrk1* and targeted *Lrrk2* mice. Floxed *Lrrk1* mice were confirmed by Southern analysis using the 5' and 3' external probes (*Figure 1D*). Targeted *Lrrk2* mice were confirmed by Southern analysis (data shown in *Figure 1—figure supplement 4*; *Nhe*I digestion followed by hybridization with the 5' external probe and *Sph*I digestion followed by hybridization with the 3' external probe). Targeted *Lrrk2* mice were then bred with *Actin-FLP* deleter mice (IMSR_JAX:005703) (*Rodríguez et al., 2000*) to generate floxed *Lrrk2* mice, which were confirmed by Southern analysis using the 5' and 3' external probes following *Nhe*I digestion (*Figure 1F*). Heterozygous *Lrrk1*^F/+ and *Lrrk2*^F/+ mice were crossed with each other to obtain homozygous single-floxed mice (*Lrrk1*^F/F and *Lrrk2*^F/F) and double-floxed mice (*Lrrk1*^F/F; *Lrrk2*^F/F).

## Generation of deleted *Lrrk1/Lrrk2* alleles and DA neuron-specific cDKO mice

In order to ensure that Cre-mediated deletion of the floxed *Lrrk1* and *Lrrk2* alleles results in null alleles, we crossed *Lrrk1*^F/F and *Lrrk2*^F/F mice with germline deleter, *CMV-Cre* mice (*B6.C-Tg(CMV-Cre)1Cgn/J*; IMSR_JAX:006054) (*Schwenk et al., 1995*), to generate *Lrrk1* deleted (Δ/Δ) mice (by removing exons 27–29) and *Lrrk2* deleted (Δ/Δ) mice (by removing the promoter region and exons 1–2) for further molecular characterization. To generate DA neuron-specific *Lrrk1/Lrrk2* cDKO mice, we used *Slc6a3-Cre* KI mice (*B6.SJL-Slc6a3tm1.1(cre)Bkmn/J*; IMSR_JAX:006660), which express Cre recombinase under the control of the endogenous *Slc6a3* promoter (*Bäckman et al., 2006*). We crossed double-floxed mice with *Slc6a3*^Cre/+ mice to generate *Lrrk* cDKO mice (*Lrrk1*^F/F; *Lrrk2*^F/F; *Slc6a3*^Cre/+). *Lrrk* cDKO and littermate control mice used in the phenotypic analysis were obtained by crossing *Lrrk1*^F/F; *Lrrk2*^F/F; *Slc6a3*^Cre/+ with *Lrrk1*^F/F; *Lrrk2*^F/F mice. We only used cDKO and control mice that carry all floxed *Lrrk1* and *Lrrk2* alleles (*Lrrk1*^F/F; *Lrrk2*^F/F) for phenotypic analysis.

## Southern analysis

For the identification and validation of the targeted and floxed *Lrrk1* alleles, we used the 5' external, 3' external, and *neo* probes. Genomic DNA from ES cells or mouse tails was digested with *Hind*III. The 5' external probe (377 bp), which is 839 bp upstream of the 5' homologous region, was PCR amplified from mouse BAC DNA (clone RP23-213J23) using primers P15 and P16. The 3' external probe (305 bp), which is 2736 bp downstream of the 3' homologous region, was PCR amplified from mouse BAC DNA (clone RP23-213J23) using primers P17 and P18. The *neo* probe (363 bp) was PCR amplified from pSoriano plasmid using primers P23 and P24. Following *Hind*III digestion, the presence of the 17.0 kb product using either the 5' or 3' external probe represents the wild-type allele, whereas the 4.8 kb (the 5' external probe) and the 12.2 kb (the 3' external probe) products represent the floxed *Lrrk1* allele. Genomic DNA digested by *Hind*III and hybridized with the *neo* probe further confirmed the wild-type and floxed alleles (no band) and the targeted *Lrrk1* allele (14.1 kb).

For the identification and validation of the targeted and floxed *Lrrk2* alleles, we used the 5' and 3' external probes as well as the *neo* probe. Genomic DNA from ES cells or mouse tails was digested with *Nhe*I or *Sph*I followed by hybridization with the 5' or 3' external probe or the *neo* probe. The 5' external probe (753 bp), which is 84 bp upstream of the 5' homologous region, was PCR amplified from mouse BAC DNA (clone RP23-526A2) using primers P19 and P20. The 3' external probe (622 bp), which is 38 bp downstream of the 3' homologous region, was PCR amplified from mouse BAC DNA (clone RP23-526A2) using primers P21 and P22. Following *Nhe*I digestion, the presence of the 11.5 kb product using either the 5' or 3' external probe represents the wild-type allele, whereas the 3.6 kb (the 5' external probe) and the 5.2 kb (the 3' external probe) products represent the floxed *Lrrk2* allele.

## PCR genotyping

Genomic PCR was performed to determine the presence of the deleted, the floxed, and/or the wild-type alleles. For *Lrrk1*, the following primers were used: P25 (5'-ATTGGTCTTTGAAGAGACAG CATCTGG, forward primer, 392 nt downstream of exon 26), P26 (5'-TTTCCCTGAGGTGGAGAAGT GACTGG, reverse primer, 567 nt downstream of exon 26), and P27 (5'-TCACGTCGTCTAAGCCTCCT , reverse primer, 1218 nt downstream of exon 29). The PCR products from P25 and P26 are 266 bp

and 405 bp, which represent the wild-type and the floxed *Lrrk1* alleles, respectively, whereas the PCR product from P25 and P27 is 583 bp, which represents the deleted *Lrrk1* allele.

For *Lrrk2*, the following primers were used: P28 (5'-CTTCCTCAGAAGTTAGGTAAACATTGAGTG, forward primer, 2069 nt upstream of exon 1), P29 (5'-CTAAGTGACACCGTGTTTCCAAAGTC, reverse primer, 1739 nt upstream of exon 1), and P30 (5'-GGAAAGTTTCACAATTGGAAAAATAAAAATAT TTACTGCAGATA, reverse primer at 2848 nt downstream of exon 2). The PCR products from P28 and P29 are 305 bp and 367 bp, representing the wild-type and the floxed *Lrrk2* alleles, respectively, whereas the PCR product from P28 and P30 is 587 bp, which represents the deleted *Lrrk2* allele.

For *Slc6a3-IRES-Cre*, the following primers were used: JKM1823 (5'-TGGCTGTTGGTGTAAAGTGG , forward primer at exon 16 and 3'-UTR), JKM1824 (5'-GGACAGGGACATGGTTGACT, reverse primer at 3'-UTR), and JKM1825 (5'-CCAAAAGACGGCAATATGGT, reverse primer at *IRES* sequence). The PCR product from JKM1823 and JKM1824 is 264 bp, which represents the wild-type allele, whereas the PCR product from JKM1823 and JKM1825 is 152 bp, which represents the *Slc6a3-IRES-Cre* KI allele.

## Northern analysis

Total RNA was isolated from brains, kidneys, or lungs using TRI reagent (T9424, MilliporeSigma) according to the manufacturer's instruction. For the *Lrrk1* northern analysis, polyA+ RNA was enriched from ~500 µg total RNA using the Poly(A)Purist MAG Kit (AM1922, Thermo Fisher) according to the manufacturer's instruction. For the *Lrrk2*, ~10 µg of total RNA was used for northern analysis. RNA was separated in formaldehyde agarose gel and transferred into Amersham Hybond-nylon membrane (RPN303N, GE Healthcare). Probes were synthesized using Prime-It II random labeling kit (#300385, Agilent) and then used for membrane hybridization at 55°C overnight.

The cDNA probe specific for *Lrrk1* exons 2–3 (383 bp) was PCR amplified using primers P31 (5'-CAGGATGAGCGTGTGTCTGCAG) and P32 (5'-CCTTCTCCTGTGAGGATTCGCTCT). The cDNA probe specific for *Lrrk1* exons 27–29 (550 bp) was PCR amplified using primers P33 (5'-CTGGCCTA CCTGCACAAGAA) and P34 (5'-CCTTCCCATCCCAGAACACC).

The cDNA probe specific for *Lrrk2* exons 1–5 (437 bp) was PCR amplified using primers P35 (5'-AGGAAGGCAAGCAGATCGAG) and P36 (5'-GGCTGAATATCTGTGCATGGC). The probe specific for *GAPDH* exons 5–7 (452 bp) was PCR amplified using primers P37 (5'-ACCACAGTCCATGCCATCAC) and P38 (5'-TCCACCACCCTGTTGCTGTA).

Hybridization was performed using α-$^{32}$P-dCTP-labeled probes specific to each gene. Specific signals were detected using autoradiography with Hyperfilm (E3018, Amersham).

## RT-PCR

Total RNA was isolated from brains, kidneys, or lungs using TRI reagent (T9424, MilliporeSigma) according to the manufacturer's instruction. Approximately 1 µg of RNA was reverse-transcribed using Superscript III (18080093, Thermo Fisher Scientific) according to the manufacturer's instructions. For RT-PCR analysis of *Lrrk1* transcripts in mice carrying the homozygous floxed or deleted alleles, we used primers P53 and P54 for exons 4–8 (714 bp), P57 and P58 for exons 11–17 (818 bp), P63 and P64 for exons 20–25 (922 bp), or P41 and P42 for exons 25–31 to confirm normal splicing of *Lrrk1* mRNA in *Lrrk1*$^{F/F}$ mice and truncated *Lrrk1* transcripts lacking exons 27–29 in Δ/Δ mice (for details, see *Figure 1—figure supplement 7*). For RT-PCR analysis of *Lrrk2* transcripts, we used primers P35 and P36 in exons 1–5 to confirm normal splicing of *Lrrk2* mRNA (437 bp) in *Lrrk2*$^{F/F}$ mice and the absence of RT-PCR products in *Lrrk2*$^{Δ/Δ}$ mice (*Figure 1—figure supplement 9*). The identity of the PCR products was confirmed by sequencing.

## Western analysis

Fresh tissues were collected and homogenized in an ice-cold stringent RIPA buffer (50 mM Tris–Cl [pH 7.6], 150 mM NaCl, 0.5 mM EDTA, 1% NP40, 0.5% sodium deoxycholate, 0.1% SDS, 1 mM PMSF supplement with protease inhibitor cocktail [P8340, Sigma], and phosphatase inhibitor cocktail [P0044, Sigma]), followed by sonication. Homogenates were centrifuged at 14,000 × *g* for 20 min at 4°C to separate supernatants (RIPA buffer-soluble fraction). An equal amount (10–40 µg per lane) of total proteins from each preparation were loaded and separated on NuPAGE gels (Invitrogen), then transferred to nitrocellulose membranes. The membranes were blocked in Intercept (TBS)

Blocking Buffer (927-60001, LI-COR) for 1 hr at room temperature and incubated at 4°C overnight with specific primary antibodies. Primary antibodies used were rabbit anti-LRRK1 (ANR-101, Alomone Lab, RRID:AB_2756700), rabbit anti-LRRK2 (ab133474, abcam, RRID:AB_2713963), and mouse anti-α-Vinculin (05-386, Millipore, RRID:AB_309711). Membranes were then incubated with dye-coupled secondary antibodies, goat anti-mouse IRdye680 (#925-68070, LI-COR, RRID:AB_2651128), or goat anti-rabbit IRdye800 (#925-32211, LI-COR, RRID:AB_2651127). Signals were quantified using the Odyssey Infrared Imaging System (LI-COR).

## Histological analysis

Mice were anesthetized with ketamine (100 mg/kg) + xylazine (10 mg/kg) + acepromazine (3 mg/kg), and transcardially perfused with phosphate-buffered saline solution (PBS, pH 7.4) containing 0.25 g/l heparin (H3149, Sigma) and 5 g/l procaine (P9879, Sigma). Brains were post-fixed in 4% formaldehyde in PBS (pH 7.4) (15710, Electron Microscopy Sciences) at 4°C overnight and then processed for paraffin embedding following standard procedures. For frozen sections, post-fixed brains were immersed in a sucrose series solution (15 and 30% sucrose in PBS) at 4°C overnight for cryoprotection, and then brains were embedded in an OCT compound (4583, Sakura). Serial coronal sections (16 μm) of paraffinized brains or frozen brains were obtained using Leica RM2235 or Leica CM1860, respectively. Coronal sections containing the SNpc or LC were selected for immunohistochemical analysis.

Histological analyses were performed as described previously (*Giaime et al., 2017*; *Huang et al., 2022*; *Kang et al., 2021*). Briefly, for DAB-derived TH-immunohistochemistry, coronal sections were deparaffinized, alcohol-dehydrated, and then subjected to permeabilization with a solution containing 0.1% Triton X-100 in TBS followed by antigen retrieval for 5 min in 10 mM sodium citrate buffer, pH 6.0. Endogenous peroxidase activity was quenched by incubating in 0.3% $H_2O_2$ in methanol. Sections were then blocked with a solution containing 5% normal goat serum (S-1000, Vector Laboratories) and 0.1% Triton X-100 in TBS for 1 hr at room temperature. After blocking, sections were incubated with the primary antibody, rabbit anti-TH (1:750, ab112, abcam, RRID:AB_297840) overnight at 4°C. Sections were washed three times in 0.1% Triton X-100 in TBS followed by 1 hr incubation with goat biotinylated anti-rabbit IgG secondary antibody (1:250, BA-1000, Vector Laboratories, RRID:AB_2313606) at room temperature and 30 min incubation with Vectastain Elite ABC reagent (PK-6100, Vector Laboratories) and then developed using chromogenic DAB substrate (SK-4100, Vector Laboratories).

For immunofluorescence staining of paraffin sections, coronal sections were deparaffinized, alcohol-dehydrated, and then subjected to permeabilization with a solution containing 0.1% Triton X-100 in PBS followed by antigen retrieval for 5 min in 10 mM sodium citrate buffer, pH 6.0, except those for cleaved-caspase-3 immunostaining that were performed antigen retrieval for 10 min. Sections were then blocked with a solution containing 10% normal goat serum and 0.1% Triton X-100 in PBS for 1 hr at room temperature. After blocking, sections were incubated with primary antibodies overnight at 4°C. The primary antibodies used were mouse anti-TH (1:50, sc-25269, Santa Cruz, RRID:AB_628422), rabbit anti-NeuN (1:400, 12943S, Cell Signaling Technology, RRID:AB_2630395), rabbit anti-cleaved caspase-3 (1:150, 9661, Cell Signaling Technology, RRID:AB_2341188), or rabbit anti-Iba1 (1:500, 019-19751, Wako, RRID:AB_839504). Sections were washed three times in 0.1% Triton X-100 in PBS followed by 1 hr incubation with fluorophore-conjugated secondary antibodies, goat anti-mouse IgG Alexa Fluor 488 (1:500, A11001, Thermo Fisher, RRID:AB_2534069), and goat anti-rabbit IgG Alexa Fluor 555 (1:500, A32732, Thermo Fisher, RRID:AB_2633281) at room temperature. Fluorescence images were taken (a stack of 2–3 confocal images spaced at 4 μm), projected with maximal intensity projection mode, and analyzed using the FV1000 confocal microscope system (Olympus).

For immunofluorescence staining of cryopreserved sections, coronal brain sections were washed with PBS to rinse out OCT and then blocked with a solution containing 5% normal goat serum and 0.1% Triton X-100 in TBS for 1 hr at room temperature. After blocking with 10% NGS, sections were incubated with primary antibodies, rabbit anti-GFP (1: 1000, ab290, abcam, RRID:AB_303395), and mouse anti-TH (1:50, sc-25269, Santa Cruz, RRID:AB_628422) overnight at 4°C. Sections were incubated with fluorophore-conjugated secondary antibodies, goat anti-rabbit IgG Alexa Fluor 488 (1:500, A11034, Thermo Fisher, RRID:AB_2576217) and goat anti-mouse IgG Alexa Fluor 555 (1:250, A21424, Thermo Fisher, RRID:AB_141780) for 1 hr at room temperature. Fluorescence images were taken and analyzed using the FV1000 confocal microscope system (Olympus).

The number of GFP+ and GFP+/TH+ cells in the SNpc of *Slc6a3*<sup>Cre/+</sup>; *Rosa26*<sup>CAG-LSL-ZsGreen1/+</sup> reporter mice was quantified using three comparable coronal sections (16 µm in thickness, spaced 320 µm apart) per brain (n = 3 brains, one hemisphere). The percentage of GFP+ DA neurons in the SNpc was obtained by dividing the sum of GPF+/TH+ neurons by the sum of total TH+ neurons quantified.

## Quantification of DA neurons in the SNpc

Quantification of TH+ DA neurons in the SNpc or LC was performed as previously described (*Giaime et al., 2017*; *Huang et al., 2022*; *Kang et al., 2021*; *Yamaguchi and Shen, 2013*). Briefly, TH+ neurons in the SNpc, which were marked based on morphological features as previously described (*Nelson et al., 1996*), were quantified in every 10th serial coronal section (16 µm in thickness) throughout the SNpc (a total of 6–9 sections, spaced 160 µm apart). Total number of TH+ cells in the SNpc was calculated as follows: [total number of TH+ DA neurons quantified in all 6–9 sections] × 10 (every 10th section sampled) × 2 (both hemispheres). Total number of TH+ noradrenergic neurons in the LC (a total of 6–9 sections, spaced 80 µm apart) was calculated as follows: [total number of TH+ noradrenergic neurons quantified in all 6–9 sections] × 5 (every 5th section sampled) × 2 (both hemispheres).

DA neuron quantification was also performed independently by another investigator, also in a genotype-blind manner, using stereological quantification of 25% of the SNpc area. The fractionator with 100 µm × 100 µm was set up in the SNpc, and TH+ DA neurons were counted using the optical dissector method with 50 µm × 50 µm sample box (25% of the total area). Total number of TH+ cells in the SNpc was calculated as follows: [total number of TH+ DA neurons quantified in sample boxes of all 6–9 sections] × 10 (every 10th section sampled) × 4 (1/4 area sampled: 50 × 50/100 × 100) × 2 (both hemispheres).

## Quantification of NeuN+ neurons, apoptotic cells, and Iba1+ microglia in the SNpc

NeuN+, active Caspase-3+, or Iba1+ cells in the SNpc, which was marked by TH immunoreactivity, were quantified using serial coronal sections (16 µm in thickness, every 10th section, a total of 6–9 sections per brain). Compared to DAB immunostaining followed by counting under the stereomicroscope, which captures TH-positive DA neurons in a single-cell layer of the brain section, immunofluorescent staining picks up more cells (e.g., TH+ or NeuN+) in multiple layers of the brain section under the confocal microscope by the maximal intensity projection mode. Thus, more TH-positive DA neurons in the SNpc were captured and quantified under confocal microscopy compared to DAB staining. The total number of NeuN+ cells in the SNpc was calculated by multiplying the [total number of NeuN+ cells in all sections counted] × 10 (every 10th section sampled) × 2 (both hemispheres). The total number of active Caspase-3+ or Iba1+ cells in the SNpc, which was marked by TH immunoreactivity, was calculated as follows: [total number of active Caspase-3+ or Iba1+ cells in all sections counted from one hemisphere] × 10 (every 10th section sampled).

## Quantification of TH+ DA terminals in the striatum

For quantification of TH immunoreactivity in the striatum, we performed immunostaining using every 10th serial coronal sections (16 µm in thickness) throughout the entire striatum (a total of 12–15 sections, spaced 160 µm apart). The images of TH immunoreactivity in the striatum were captured under 2× objective lens (Olympus BX40, 8-bit RGB camera) using identical exposure time, sensitivity, brightness, contrast, and gamma (CellSens Entry Software) and then analyzed using the Fiji version of ImageJ, and the optical density was determined as previously described (*Huang et al., 2022*; *Ventruto et al., 1975*; *Schindelin et al., 2012*). The mean value of TH immunoreactivity in the striatum of control mice of each age group was set as 100%.

## Quantitative EM analysis

The collection and quantification of the EM images were performed as described previously (*Giaime et al., 2017*; *Huang et al., 2022*). Mice were perfused with PBS containing 0.25 g/l heparin and 5 g/l procaine followed by a fixative solution containing 2.5% paraformaldehyde and 2.5% glutaraldehyde in 0.1 M sodium cacodylate buffer (pH 7.4) (#1549, Electron Microscopy Sciences). Brains were dissected and post-fixed overnight in a fixative solution at 4°C. The dissected tissues were trimmed

to 1–2 mm$^3$ cubes followed by osmication and uranyl acetate staining, dehydration in graded alcohol, and embedded in TABB 812 Resin (Marivac Ltd) at the Harvard Medical School EM facility. 0.5 µm sections were stained with toluidine blue and viewed under the light stereomicroscope (Nikon Eclips E600) to find the SNpc area for EM viewing. Then adjacent sections were cut with 80 nm in thickness with the Leica Ultracut S microtome, picked up on formvar-carbon-coated slot copper grids, stained with 0.2% lead citrate, and viewed and imaged under the JEOL 1200× electron microscope. A minimum of 10 micrographs containing the entire cell body in the SNpc area were analyzed for each brain. The image was analyzed using the Fiji version of ImageJ. The number of electron-dense autophagic and lysosomal vacuoles (>0.5 µm in diameter) in individual neuronal profiles was quantified. We previously calculated the diameter of electron-dense autophagic/lysosomal vacuoles by measuring the longest side manually (*Giaime et al., 2017*; *Huang et al., 2022*). In the current study, we used Feret's diameter (*Walton, 1948*) to accurately measure the longest distance between any of the two points in the electron-dense autophagic/lysosomal vacuoles, resulting in a higher number of electron-dense vacuoles quantified. The number and the area of electron-dense vacuoles (>0.5 µm in diameter) in individual neuronal profiles were quantified, and the average number or area of vacuoles per mouse was calculated. Experiments were done in a genotype-blind manner (after scarifying mice, the brain samples were coded and sent to the Harvard Medical School EM Core, where the images were captured, and the vacuoles were quantified by another independent experimenter).

## Behavioral analysis

Naive *Lrrk* cDKO mice and littermate controls at 10 and 22 months of age were used. Mice were acclimated in the behavior facility for a minimum of 7 d and were then individually handled daily for five consecutive days before testing. Mice were coded, so the experimenter was unaware of their genotypes until the data analysis was complete. In the beam walk test, a Plexiglas beam of 100 cm in length (Plastic Zone), 20 mm or 10 mm in width, was raised 60 cm above a table, and safety bedding was placed under the beam to avoid any harm in case of falls. Mice were placed onto the starting point of the beam in bright light, and the time (in seconds) to reach their home cage on the other darker side of the beam (~80 cm in distance) as well as the hindpaw slips (number of hindlimb errors) was recorded. Mice were trained three trials per day for two consecutive days to traverse the 20 mm beam (without the wire mesh) to their home cage. On the test day, mice were trained further with two additional trials on the 20 mm beam (without the wire mesh). Mice were then tested in two successive trials on the 20 mm beam (with the wire mesh) followed by two consecutive test trials on the 10 mm beam (with the wire mesh). All test trials were videotaped, and the travel time and the number of hindlimb errors were recorded. Between-trials mice were placed in the home cage for 2 min to rest. Mice that fell off the beam during both trials or stalled on the beam for more than 120 s during the test were excluded.

In the pole test, mice were placed on the top of the pole (60 cm in height, 10 mm in diameter) with their head facing upward, and the base of the pole was placed in the home cage. Mice were trained three trials per day for 2 d to traverse the pole to the cage floor and were further trained two trials before testing on the test day. Mice were then tested for two trials, and the time to turn around (turning time) and the time to descend the pole (descending time) were recorded. Between trials, mice were placed in the home cage for 2 min to rest. Mice that stalled on the top of the pole for more than 120 s were excluded.

## Experimental design and statistical analysis

All experiments were performed in a genotype-blind manner with the exception of molecular analysis (Southern, northern, RT-PCR, and western). All statistical analyses were performed using Prism 9 (GraphPad Software) or Excel (Microsoft). All data are presented as the means ± SEM. The exact sample size of each experiment is indicated in the figure or the legend. The slight difference in sample size among various histological analyses (e.g., at 24M, 11 control and 8 cDKO brains analyzed for TH+ cells but 9 control and 8 cDKO analyzed for NeuN+, Caspase-3+, Iba1+ cells in the SNpc) is due to the paraffin blocks used for quantification (each paraffin block contains seven brains of mixed genotypes; for details, see source data files) or specific brain sections being damaged, thus, excluded in the analysis.

Statistical analyses were conducted using an unpaired two-tailed Student's *t*-test for the comparison of a given variable in two genotypes or two-way ANOVA followed by Bonferroni's post hoc comparisons for the comparison of more than two conditions. Statistical outliers were identified and excluded using the ROUT method with 1% the maximum desired false discovery rate developed by Prism, and the only statistical outliers of the current study were identified in the behavioral analysis and are marked in *Figure 9—source data 1*. All statistical analyses were performed on data from ≥4 mouse brains per genotype per age group, and experiments were performed and repeated on different days, often by independent investigators. Statistical significance is indicated as *p<0.05, **p<0.01, ***p<0.001, ****p<0.0001, and values not significantly different are often not noted.

## Materials availability

Requests for materials generated in the current study such as the floxed *Lrrk1* and *Lrrk2* mice should be directed to and will be fulfilled by Dr. Jie Shen (jshen@bwh.harvard.edu).

## Acknowledgements

We thank H Zhao for technical assistance, Y Yuan for organizing experimental data, and Shen lab members for discussion. We are grateful to M Ericsson at the Harvard Medical School Electron Microscopy Core for her expert assistance. This work was supported by a grant from the NIH (R37NS071251 to JS).

## Additional information

### Funding

| Funder | Grant reference number | Author |
|---|---|---|
| National Institutes of Health | R37NS071251 | Jie Shen |
| National Institutes of Health | P50NS094733 | Jie Shen |

The funders had no role in study design, data collection and interpretation, or the decision to submit the work for publication.

### Author contributions

Jongkyun Kang, Data curation, Validation, Investigation, Methodology, Writing – original draft, Writing – review and editing, Western analysis of deleted LRRK1 and LRRK2 mice as well as cDKO mice; quantitative histological and EM analyses of the SNpc, including independent quantification of TH+ neurons in the SNpc and quantification of TH+ terminals in the striatum; Guodong Huang, Data curation, Investigation, Methodology, Writing – original draft, Northern analysis and RT-PCR followed by sequencing of floxed and deleted LRRK1 and LRRK2 mice; quantitative histological and EM analyses of the SNpc, including independent quantification of TH+ neurons in the SNpc and quantification of TH+ terminals in the striatum; independent behavioral validation; Long Ma, Data curation, Investigation, Methodology, Generated the targeting vectors and performed ES cell experiments and Southern blotting for LRRK1; Youren Tong, Data curation, Investigation, Methodology, Generated the targeting vectors and performed ES cell experiments and Southern blotting for LRRK2; Anu Shahapal, Data curation, Investigation, Methodology, Writing – review and editing, Performed behavioral analysis; Phoenix Chen, Data curation, Investigation, Methodology, Performed the GFP reporter analysis of DAT-Cre mice in the SNpc and the LC; Jie Shen, Conceptualization, Supervision, Funding acquisition, Writing – original draft, Project administration, Writing – review and editing

### Author ORCIDs

Jie Shen http://orcid.org/0000-0003-1160-7118

## Ethics

All animal use was approved by the IACUC committees of Harvard Medical School and Brigham and Women's Hospital (#2016N000120), and conformed to the USDA Animal Welfare Act, PHS Policy on Humane Care and Use of Laboratory Animals, the "ILAR Guide for the Care and Use of Laboratory Animals" and other applicable laws and regulations. Mice were housed in constant humidity- and temperature-controlled rooms and maintained on a 12 hr light/dark cycle and were given standard rodent chow and water.

Reviewer #1 (Public Review): https://doi.org/10.7554/eLife.92673.4.sa1
Reviewer #2 (Public Review): https://doi.org/10.7554/eLife.92673.4.sa2
Reviewer #3 (Public Review): https://doi.org/10.7554/eLife.92673.4.sa3
Author response https://doi.org/10.7554/eLife.92673.4.sa4

---

# Additional files

## Supplementary files

- MDAR checklist
- Supplementary file 1. Generation of the Lrrk1 targeting vector.
- Supplementary file 2. Generation of the Lrrk2 targeting vector.

## Data availability

All data generated or analysed during this study are included in the manuscript and supporting files; source data files have been provided for key findings shown in Figures 1, 3, 4, 8, and 9.

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

# Appendix 1

**Appendix 1—table 1.** Oligonucleotides.

**Oligonucleotides**

---

*Lrrk1* PCR forward primer for left arm amplification: P1
5'-GACAT<u>CCGCGG</u>CACCATGTGAGTGGCAGCTGTGGTGAGAAC-3'
*Sac*II sequence is underlined

---

*Lrrk1* PCR reverse primer for left arm amplification: P2
5'-GACAT<u>GCGGCCGCAAGCTT</u>TTTAATAGCCGTTCTTTCTTAGAGAAGGCAG-3'
*Not*I and *Hind*III sequences are underlined

---

*Lrrk1* PCR forward primer for left-middle arm amplification: P3
5'-CCAGTCACTTCTCCACCTCAGGGAAAATGG-3'

---

*Lrrk1* PCR reverse primer for left-middle arm amplification: P4
5'-CCTTGT<u>GGTACC</u>CGGACCTTCTATCACCTTTATCC-3'
*Kpn*I sequence is underlined

---

*Lrrk1* PCR forward primer for right-middle arm amplification: P5
5'-GGTCCG<u>GGTACC</u>ACAAGGTGCTGGTTAAGTGCC-3'
*Kpn*I sequence is underlined

---

*Lrrk1* PCR reverse primer for right-middle arm amplification: P6
5'-AGCAGACCTCTTGCCTTCTACTACTGACTG-3'

---

*Lrrk1* PCR forward primer for right arm amplification: P7
5'- GACAT<u>GTCGAC</u>GGATCCGTAGGGAAGACCCACTAGGAGGAAGAAAG-3'
*Sal*I sequence is underlined

---

*Lrrk1* PCR reverse primer for right arm amplification: P8
5'-GACAT<u>AAGCTT</u>TGGTACCTTTCTAAAGGCAGCATTTTGCTTGC-3'
*Hind*III sequence is underlined

---

*Lrrk1* PCR forward primer for 5' Southern probe: P15
5'-CAGAATACCCCATGCTGGGGAATTGC-3'

---

*Lrrk1* PCR reverse primer for 5' Southern probe: P16
5'-CCGTTTCTAGGATTCTAATTTTTC-3'

---

*Lrrk1* PCR forward primer for 3' Southern probe: P17
5'-AACTCCTTCCTGGTGCTGGCAGGCCTGGCTG-3'

---

*Lrrk1* PCR reverse primer for 3' Southern probe: P18
5'-ACAAGTGACGTGACCATGGACGGAGCTGCG-3'

---

*Neo* PCR forward primer for Southern probe: P23
5'-ATTCGGCTATGACTGGGCAC-3'

---

*Neo* PCR reverse primer for Southern probe: P24
5'-GACCACCAAGCGAAACATCG-3'

---

*Lrrk1* PCR forward primer for exons 2–3 northern probe: P31
5'-CAGGATGAGCGTGTGTCTGCAG-3'

---

*Lrrk1* PCR reverse primer for exons 2–3 northern probe: P32
5'-CCTTCTCCTGTGAGGATTCGCTCT-3'

---

*Lrrk1* PCR forward primer for exons 27–29 northern probe: P33
5'-CTGGCCTACCTGCACAAGAA-3'

---

*Lrrk1* PCR reverse primer for exons 27–29 northern probe: P34
5'-CCTTCCCATCCCAGAACACC-3'

---

*GADPH* PCR forward primer for northern probe: P37
5'-ACCACAGTCCATGCCATCAC-3'

---

*GADPH* PCR reverse primer for northern probe: P38
5'-TCCACCACCCTGTTGCTGTA-3'

---

*Appendix 1—table 1 Continued on next page*

*Appendix 1—table 1 Continued*

**Oligonucleotides**

*Lrrk1* RT primer in exon 32: P47

5'-GGCTCAGGTCATGCTCAGTT-3'

*Lrrk1* PCR forward primer for exons 4–8 RT-PCR: P53
5'-TTTTGGACACGCCGAAGTAGT-3'

*Lrrk1* PCR reverse primer for exons 4–8 RT-PCR: P54
5'-AGCCGCTCCAGGTAGTTTTT-3'

*Lrrk1* PCR forward primer for exons 11–17 RT-PCR: P57
5'-GGACCTCTCCAGAAACCAGC-3'

*Lrrk1* PCR reverse primer for exons 11–17 RT-PCR: P58
5'-GCAGGGTTGCTATCCTCTCC-3'

*Lrrk1* PCR forward primer for exons 20–25 RT-PCR: P63
5'-GCGGTCAGTGGCAAAGAATG-3'

*Lrrk1* PCR reverse primer for exons 20–25 RT-PCR: P64
5'-AATGCTGTTCTCACCCTCCG-3'

*Lrrk1* PCR forward primer for exons 25–31 RT-PCR: P41
5'-GAATTCTGCTAATGCCCCAGC-3'

*Lrrk1* PCR reverse primer for exons 25–31 RT-PCR: P42
5'-AGGCTGTAGATATAGATTTTCTGGT-3'

*Lrrk2* PCR forward primer for left arm amplification: P9
5'-GAACACACAAGGCTATGGCTATTGTC-3'

*Lrrk2* PCR reverse primer for left arm amplification: P10
5'-GTAGGACTATCATCCACCTGTAGGACTCC-3'

*loxP* site introducing oligo for *Lrrk2*: P39
5'-<u>GATCC</u>ATAACTTCGTATAATGTATGCTATACGAAGTTAT<u>GCTAGC</u>A-3'
*BamH*I and *Nhe*I sequences are underlined

*loxP* site introducing oligo for *Lrrk2*: P40
5'-<u>CTAGTGCTAGC</u>ATAACTTCGTATAGCATACATTATACGAAGTTATG-3'
*Spe*I and *Nhe*I sequences are underlined

*Lrrk2* PCR forward primer for middle arm amplification: P11
5'-GTCCTACAGGTGGAT<u>TCTAGA</u>CCTACAAGG-3'
*Xba*I sequence is underlined

*Lrrk2* PCR reverse primer for middle arm amplification: P12
5'-GAC<u>GCGGCCGCG</u>TTTAGTA<u>GCTAGC</u>ATGACTATGAAGGAG-3'
*Not*I and *Nhe*I sequences are underlined

*Lrrk2* PCR forward primer for right arm amplification: P13
5'-GCACTTGAGTCTTAATCTTGGGCAC-3'

*Lrrk2* PCR reverse primer for right arm amplification: P14
5'-CATTCGAGCAGCTAAGCCTGTAATC-3'

*Lrrk2* PCR forward primer for 5' Southern probe: P19
5'-CATGGGAGAGAGGGTTTCTCACTTACT-3'

*Lrrk2* PCR reverse primer for 5' Southern probe: P20
5'-CTTGGACAGCATTGTCAGCCTAGAC-3'

*Lrrk2* PCR forward primer for 3' Southern probe: P21
5'-GCCAAGCAGTTATTGATGCTGTAGC-3'

*Lrrk2* PCR reverse primer for 3' Southern probe: P22
5'-CGAGCTGTAAGATGAGCTGGGTACT-3'

*Lrrk2* PCR forward primer for northern probe: P35
5'-AGGAAGGCAAGCAGATCGAG-3'

*Appendix 1—table 1 Continued on next page*

*Appendix 1—table 1 Continued*

**Oligonucleotides**

---

*Lrrk2* PCR reverse primer for northern probe: P36
5'-GGCTGAATATCTGTGCATGGC-3'

---

*Lrrk2* RT primer in exons 51: P93
5'-TCGTGTGGAAGATTGAGGTCC-3'

---

*Lrrk1* PCR forward primer for genotyping: P25
5'-ATTGGTCTTTGAAGAGACAGCATCTGG-3'

---

*Lrrk1* PCR reverse primer for genotyping: P26
5'-TTTCCCTGAGGTGGAGAAGTGACTGG-3'

---

*Lrrk1* PCR reverse primer for genotyping: P27
5'-TCACGTCGTCTAAGCCTCCT-3'

---

*Lrrk1* PCR forward primer for genotyping: P28
5'-CTTCCTCAGAAGTTAGGTAAACATTG AGTG-3'

---

*Lrrk1* PCR reverse primer for genotyping: P29
5'-CTAAGTGACACCGTGTTTCCAAAGTC-3'

---

*Lrrk1* PCR reverse primer for genotyping: P30
5'-GGAAAGTTTCACAATTGGAAAAATAAAAATATTTACTGCAGATA-3'

---

*Slc6a3-Cre* PCR forward primer for genotyping: JKM1823
5'-TGGCTGTTGGTGTAAGTGG-3'

---

*Slc6a3-Cre* PCR reverse primer for genotyping: JKM1824
5'-GGACAGGGACATGGTTGACT-3'

---

*Slc6a3-Cre* PCR reverse primer for genotyping: JKM1825
5'-CCAAAGACGGCAATATGGT-3'

---

