## [Editor Report · eLife assessment]

This current revision builds on observations in validated conditional double KO (cDKO) mice for LRRK1 and LRRK2 that will be **useful** for the field, given that LRRK2 is widely expressed in the brain and periphery, and many divergent phenotypes have been attributed previously to LRRK2 expression. The article presents **solid** data demonstrating that it is the loss of LRRK1 and LRRK2 expression within the SNpc DA cells that is not well tolerated as it was previously unclear from past work whether neurodegeneration in the LRRK double knock out (DKO) was cell autonomous or the result of loss of LRRK1/LRRK2 expression in other types of cells. Future studies may pursue the biochemical mechanisms underlying the reason for the apoptotic cells noted in this study, as here, the LRRK1/LRRK2 KO mice did not replicate the dramatic increase in autophagic vacuole numbers previously noted in the germline global LRRK1/LRRK2 KO mice.

---

## [Referee Report · Reviewer #1 (Public Review)]

Summary:

This is an important work showing that loss of LRRK function causes late-onset dopaminergic neurodegeneration in a cell-autonomous manner. One of the LRRK members, LRRK2, is of significant translational importance as mutations in LRRK2 cause late-onset autosomal dominant Parkinson's disease (PD). While many in the field assume that LRRK2 mutant causes PD via increased LRRK2 activity (i.e., kinase activity), it is not a settled issue as not all disease-causing mutant LRRK2 exhibits increased activity. Further, while LRRK2 inhibitors are under clinical trials for PD, the consequence of chronic, long-term LRRK2 inhibition is unknown. Thus, studies evaluating the long-term impact of LRRK deficit have important translational implications. Moreover, because LRRK proteins, particularly LRRK2, are known to modulate immune response and intracellular membrane trafficking, the study's results and the reagents will be valuable for others interested in LRRK function.

Strengths:

This report describes a mouse model where LRRK1 and LRRK2 genes are conditionally deleted in dopaminergic neurons. Previously, this group showed that while loss of LRRK2 expression does not cause brain phenotype, loss of both LRRK1 and LRRK2 causes a later onset, progressive degeneration of catecholaminergic neurons, dopaminergic (DAergic) neurons in the substantia nigra (SN) and noradrenergic neurons in the Locus Coeruleus (LC). However, because LRRK genes are widely expressed with some peripheral phenotypes, it was unknown if the neurodegeneration in LRRK double Knock Out (DKO) was cell autonomous. To rigorously test this question, the authors generated a double conditional KO allele where both LRRK1 and LRRK2 genes were targeted to contain loxP sites. This was beyond what is usually required as most investigators might just have combined one KO allele with another floxed allele. The authors provide a rigorous validation showing that the Driver (DAT-Cre) is expressed in most DAergic neurons in SN and that LRRK levels are decreased selectively in the ventral midbrain. Using these mice, the authors show that the number of DA neurons is average at 15 but significantly decreased at 20 months of age. Moreover, the authors show that the number of apoptotic neurons is increased by ~2X in aged SN, demonstrating increased ongoing cell death and activated microglia. The degeneration is limited to DA neurons as LC neurons are not lost, and this population does not express DAT. Overall, the mouse genetics and experimental analysis were performed rigorously, and the results were statistically sound and compelling.

Weakness:

I only have a few minor comments. First, in PD and other degenerative conditions, axon and terminal loss occur prior to cell bodies. It might be beneficial to show the status of DAergic markers in the striatum. Second, previous studies indicate that very little, if any, LRRK1 is expressed in SN DAergic neurons. This also the case with the Allen Brain Atlas profile. Thus, the authors should discuss the discrepancy, as they imply significant LRRK1 expression in DA neurons.

Revision:

I appreciate the authors revising the manuscript with additional data that have clarified my prior comments. They now show that TH levels in the striatum decrease with SNpc neurons. Further, while there is some disagreement regarding the expression LRRK1 in SNpc, the authors provide a convincing case that LRRK1 function is important in SNpc DA neurons.

---

## [Referee Report · Reviewer #2 (Public Review)]

Summary:

In this manuscript, Shen and collaborators described the generation of conditional double knockout (cDKO) mice lacking LRRK1 and LRRK2 selectively in DAT-positive dopaminergic neurons. The Authors asked whether selective deletion of both LRRK isoforms could lead to a Parkinsonian phenotype, as previously reported by the same group in germline double LRRK1 and LRRK2 knockout mice (PMID: 29056298). Indeed, cDKO mice developed a late reduction of TH+ neurons in SNpc that partially correlated with the reduction of NeuN+ cells. This was associated with increased apoptotic cell and microglial cell numbers in SNpc. Unlike the constitutive DKO mice described earlier, cDKO mice did not replicate the dramatic increase in autophagic vacuole numbers. The study supports the authors' hypothesis that loss of function rather than gain of function of LRRK2 leads to Parkinson's Disease.

Strengths:

For the first time, the study described a model in which both the Parkinson's disease-associated gene LRRK2 and its homolog LRRK1 are deleted selectively in dopaminergic neurons. This offers a new tool to understand the physiopathological role of LRRK2 and the compensating role of LRRK1 in modulating dopaminergic cell function.

Weaknesses:

The model has no construct validity since loss of function mutations of LRRK2 are well tolerated in humans and do not lead to Parkinson's disease. The evidence of a Parkinsonian phenotype in these conditional knockout mice is limited and should be considered preliminary.

---

## [Referee Report · Reviewer #3 (Public Review)]

Kang, Huang, and colleagues have provided new data to address concerns regarding confirmation of LRRK1 and LRRK2 deletion in their mouse model and the functional impact of the modest loss of TH+ neurons observed in the substantia nigra of their double KO mice. In the revised manuscript, the new data around the characterization of the germline-deleted LRRK1 and LRRK2 mice add confidence that LRRK1 and LRRK2 can be deleted using the genetic approach. They have also added new text to the discussion to try and address some of the comments and questions raised regarding how LRRK1/2 loss may impact cell survival and the implications of this work for PD-linked variants in LRRK2 and therapeutic approaches targeting LRRK2. The new data provides additional support for the author's claims.

---

## [Author Response]

The following is the authors’ response to the previous reviews.

**Public Reviews:**

**Reviewer #3 (Public Review):**
Kang, Huang, and colleagues have provided new data to address concerns regarding confirmation of LRRK1 and LRRK2 deletion in their mouse model and the functional impact of the modest loss of TH+ neurons observed in the substantia nigra of their double KO mice. In the revised manuscript, the new data around the characterization of the germline-deleted LRRK1 and LRRK2 mice add confidence that LRRK1 and LRRK2 can be deleted using the genetic approach. They have also added new text to the discussion to try and address some of the comments and questions raised regarding how LRRK1/2 loss may impact cell survival and the implications of this work for PD-linked variants in LRRK2 and therapeutic approaches targeting LRRK2.The new data provides additional support for the author's claims. I have provided below some suggestions for clarification/additions to the text that can be addressed without additional experiments.(1) The authors added additional text highlighting that more studies are warranted in mice where LRRK1/2 are deleted in other CNS cell types (microglia/astrocytes) to understand cell extrinsic drivers of the autophagy deficits observed in their previous work. It still remains unclear how loss of LRRK1/2 leads to increased apoptosis and gliosis in dopaminergic neurons in a cell-intrinsic manner, and, as suggested in the original review, it would be helpful to add some text to the discussion speculating on potential mechanisms by which this might occur.(2) Revisions have been made to the discussion to clarify their rationale around how variants in LRRK2 associated with PD may be loss-of-function to support the relevance of this mouse model to phenotypes observed in PD. However, as written, the argument that PD-linked variants are loss-offunction is based on the fact that the double KO mice have a mild loss of TH+ neurons while the transgenic mice overexpressing PD-linked LRRK2 variants often do not and that early characterization of kinase activity was done in vitro are relatively weak. Given that the majority of evidence generated by many labs in the field supports a gain-of-function mechanism, the discussion should be further tempered to better highlight the uncertainty around this (rather than strongly arguing for a loss-offunction effect). This could include the mention of increased Rab phosphorylation observed in cellular and animal models and opposing consequences on lysosomal function observed in cellular studies in KO and pathogenic variant expressing cells. Further, a reference to the Whiffen et al. 2020 paper mentioned by another reviewer should be included in the discussion for completeness.

We thank the reviewer for the comments. The discussion has been further revised and expanded to explain the cell extrinsic microglial response to pathophysiological changes in DA neurons of cDKO mice and propose future studies of single-cell RNA-sequencing to identify molecular changes within DA neurons of cDKO mice that may drive their apoptotic death during aging.

We also added paragraphs summarizing existing experimental evidence for the toxic gain-of-function mechanism (biochemical data of increased kinase activity but the lack of evidence for the elevated pRabs and the altered pLRRK2 driving dopaminergic neurodegeneration) and for the loss-of-function mechanism (genetic data of relevant physiological roles) as well as the relationships between LRRK1 and LRRK2 (functional homologues sharing functional domains and overlapping roles in dopaminergic neuron survival) and how dominantly inherited missense mutations can confer a loss of function mechanism (impairing its function in cis and inhibiting wild-type protein function in trans). We also provided a brief summary and discussion of the Whiffen et al. 2020 paper.